



# Modelling study of transformations of the exchange flows along the Strait of Gibraltar

Antonio Sanchez-Roman[1], Gabriel Jorda[1], Gianmaria Sannino[2] and Damia Gomis[1]

[1] IMEDEA (UIB-CSIC), Mallorca, Spain

[2] Energy and Environmental Modeling Unit, ENEA, Italy

*Correspondence to*: Antonio Sánchez-Román (asanchez@imedea.uib-csic.es)

**Abstract.** Vertical transfers of heat, salt and mass between the inflowing and outflowing layers at the Strait of Gibraltar are explored basing on the outputs of a three-dimensional, fully non-linear numerical model. The model covers the entire Mediterranean basin and has a very high spatial resolution around the Strait (1/200°). Another distinctive feature of the

model is that it includes a realistic barotropic tidal forcing (diurnal and semidiurnal), in addition to atmospheric pressure and heat and water surface fluxes. The results show a significant transformation of the properties of the inflowing and outflowing water masses along their path through the Strait. This transformation is mainly induced by the recirculation of water, and therefore of heat and salt, between the inflowing and outflowing layers. The underlying process seems to be the hydraulic control acting at the Espartel section, Camarinal Sill and Tarifa Narrows, which limits the amount of water than can cross the

sections and forces a vertical recirculation. This results in a complex spatio-temporal pattern of vertical transfers, with the sign of the net vertical transfer being opposite in each side of Camarinal Sill. Conversely, the mixing seems to have little influence on the heat and salt exchanged between layers (~2-10% of advected heat/salt). Therefore, the main point of our work is that most of the transformation of water properties along the Strait is induced by the vertical advection of heat and salt and not by vertical mixing. A simple relationship between the net flux and the vertical transfers of water, heat and salt is

also proposed. This relationship could be used for the fine tuning of coarse resolution model parameterizations in the Strait.

## 1 Introduction

The Strait of Gibraltar is a narrow and shallow channel with a length of about 60 km and a mean width of 20 km that presents a complex system of contractions and submarine sills (see Fig. 1). Between Gibraltar and Ceuta the channel is about 25 km wide and 800-900 m deep; west of this section the Strait narrows towards a minimum cross section of about 14 km

called Tarifa Narrows (TN hereinafter). To the west the bottom abruptly rises, reaching the minimum depth of the whole Strait (290 m) at Camarinal Sill (CS hereinafter). More to the west, the cross-section divides into two channels: a northern channel with a maximum depth of 250 m and a southern channel with a maximum depth of 360 m that is actually a relative minimum depth for the main along-strait channel in the western part of the Strait. This topographic point, called Espartel Sill (ES hereinafter) represents the last topographic constrain for the Mediterranean Outflow Water (MOW hereinafter)

(Sanchez-Roman et al., 2009).



As the only effective connection between the Mediterranean Sea and the open ocean, the Strait of Gibraltar plays a key role in the budgets of water, heat and salt of the Mediterranean basin (Soto-Navarro et al., 2010). Buoyancy loses over the Mediterranean Sea, as a consequence of the excess of evaporation ($E$) over precipitation ($P$) and river discharge ($R$) lead to a two-layer baroclinic exchange in the Strait (Bryden and Kinder, 1991): about 0.8 Sv (1 Sv = $10^6$ m$^3$s$^{-1}$) of cold and salty ($S_M$

≈ 38.4) MOW flows towards the Atlantic Ocean in the lower layer (Sanchez-Roman et al., 2009), and a slightly higher volume rate of fresh ($S_A ≈ 36.2$) and warm Atlantic Water (AW) spreads into the Mediterranean basin in the surface layer. A long term net inflow of the order of 0.05 Sv is necessary to balance the water deficit of the Mediterranean Sea (Soto-Navarro et al., 2010).

The subinertial flow through the Strait is not steady. It fluctuates at different timescales ranging from seasonal and inter-

annual variability (see e.g. Garcia-Lafuente et al., 2007) to intra-seasonal changes driven by winds and, mainly, by atmospheric pressure differences between the Atlantic Ocean and the Mediterranean Sea (see e.g. Sanchez-Roman et al., 2012). The flow also shows strong diurnal and semidiurnal variations due to tidal currents, which interact with the topography of the Strait and mask the underlying mean baroclinic exchange during important parts of the tidal cycle. Moreover, the magnitude and hydrological properties of the exchange flow strongly depend on the physical configuration of

the Strait (Bryden and Stommel, 1984), which submits the water exchange to hydraulic control. Depending on the number and location of the hydraulic controls, two states for the Strait dynamics are possible: a first case is the submaximal exchange regime, which occurs when the flow is only controlled either at CS or at ES. A second case is the maximal exchange, in which the flow is also controlled at TN. The two regimes have different implications for property fluxes, response time, and other physical characteristics of the coupled circulation in the Strait and in the Mediterranean Sea

(Sannino et al., 2009).

Bray et al. (1995) demonstrated that the inflowing and outflowing layers are not isolated from each other. Instead, the two layers are strongly influenced by vertical transfers and mixing between Mediterranean and Atlantic waters that result in vertical displacements of the mean interface and in changes in the hydrological properties of the waters masses throughout the Strait. According to these authors, the water mass characteristics of the exchange flows at any point of the Strait of

Gibraltar can be readily described in terms of a mixture of three principal water types stemming from the neighboring eastern North Atlantic Ocean and western Mediterranean Sea: Surface Atlantic Water (SAW), North Atlantic Central Water (NACW) and Mediterranean Water (MW), which is itself a mixture of Levantine Intermediate Water (LIW) and Western Mediterranean Deep Water (WMDW). More recently, Millot (2009, 2014) has proposed another concept for the Mediterranean Outflow, this being composed by two additional water types: Western Intermediate Water (WIW), formed in

the western basin of the Mediterranean, and Tyrrhenian Dense Water (TDW), formed by the cascading of deep Eastern Mediterranean Waters from the Channel of Sicily and mixing with Western Mediterranean resident waters.

In a two-layer framework, changes in the mixing between the incoming and outgoing waters along the Strait imply that waters entering the Mediterranean will have different temperature and salinity properties and hence different buoyancy properties. Since this inflow will be finally transformed into intermediate and/or deep water along the Mediterranean basin,





the changes along the Strait will have an impact on the thermohaline circulation of the whole basin and hence on the evolution of the Mediterranean. In an attempt to investigate this, Bray et al. (1995) analysed the exchange through the Strait by using hydrographic data and a non-tidal conceptual model consisting of an upper layer of Atlantic water (salinity less than 36.5 on average); a lower layer of Mediterranean water (salinity higher than 38.2 on average); and an interfacial layer in

between, whose properties change gradually because of the mixing. They showed that the interfacial layer presents a different behavior throughout the Strait, flowing towards the Atlantic Ocean west of CS, and toward the Mediterranean Sea east of CS. These results were corroborated by Garcia-Lafuente et al. (2000), who found that the mean surface of null velocity at the eastern entrance of the Strait is located in the lower portion of the Interfacial layer and coincides with the material surface of $S = 37.9$, thus enhancing the Atlantic inflow (fast-flowing or active layer) due to the transfer of

Mediterranean water from the lower layer (slowly flowing or passive layer). More recently, Garcia-Lafuente et al. (2013) revisited their work to investigate the dynamics of the interface mixing layer at tidal time-scales, though the model used in that latter study did not include neither the wind stress at the sea surface nor the remote forcing driven by the atmospheric pressure variations over the Mediterranean, which is able to distort the periodic tidal pattern (Vazquez et al., 2008). At the other side of the Strait, Sanchez-Roman at al. (2009) reported that the null-velocity surface at ES coincides with the surface

of $S = 36.9$, which is located in the upper portion of the Interfacial layer. There, the outflowing Mediterranean waters play the role of the active layer and takes water from the Atlantic inflow (which acts as the passive one) thus enhancing the MOW (Sanchez-Roman et al., 2012).

All these previous works discriminated between Atlantic and Mediterranean waters using a given salinity surface as separation and investigated the vertical transfers either focusing on a single section within the Strait or by using simplified

models. This makes that a quantification of the vertical transfer of properties between the inflowing and outflowing waters separated using strictly a velocity criterion is still lacking. To fill this gap, we present a numerical study on the transformations that waters suffer along their path through the Strait of Gibraltar, with emphasis on the vertical transfer of water, heat and salt between incoming and outgoing layers. Namely we use a realistic three-dimensional and fully nonlinear hydrostatic model with very high resolution in the Strait area. As a novelty with respect to previous numerical studies in the

region, the model is not only forced by realistic barotropic tides, but also by the atmospheric pressure and heat & water fluxes at the sea surface. The very high resolution and the inclusion of all the forcing factors allow a realistic representation of the exchanges at the Strait. Furthermore, we use the results of the numerical simulations to propose a simple relationship between the net water transport and the vertical transfers of water, heat and salt. This could be used for the fine tuning of mixing parameterizations in models that do not include tides and/or have a coarse resolution and therefore are not able to

explicitly simulate the water mass transformations along the Strait.

The characteristics of the numerical model and the methodology followed to compute the horizontal and vertical transports are described in Section 2. Model results are presented in Section 3 and are discussed in Section 4. Finally, conclusions are outlined in Section 5.



## 2 Data and methods

### 2.1 Numerical model and simulations

The numerical model used for this study is based on the three-dimensional z-coordinate Massachusetts Institute of Technology general circulation model (MITgcm), which was adapted to the Mediterranean region. The model solves the

fully nonlinear hydrostatic Navier–Stokes equations under the Boussinesq approximation for an incompressible fluid with a spatial finite-volume discretization on a curvilinear computational grid (Sanchez-Garrido et al., 2013). The model formulation is described in detail by Marshall et al. (1997a, 1997b) and its source code and documentation are available at the following web site (http://mitgcm.org/sealion /online_documents/node2.html).

The model uses a curvilinear orthogonal grid covering the entire Mediterranean Sea and part of the Atlantic Ocean, including

the Gulf of Cadiz at its western boundary (Sannino et al., 2015). The grid has a nonuniform horizontal spacing: over most of the model domain it is 1/16° x 1/16°, but in between the Alboran Sea and the Gulf of Cadiz, following the recommendation of Sannino et al. (2014), it increases up to a maximum value of 1/200° (~500 m) at the Strait of Gibraltar (Fig. 1b). In the vertical the grid has 72 unevenly spaced z-level in order to adequately resolve the dynamics of the different overlying water masses in the Mediterranean. The layer thickness ranges from 3 m at the sea surface to 300 m at the bottom. The partial cell

formulation is used for the near-bottom level. According to Sannino et al., (2015), the model bathymetry was obtained by a merging procedure that involved three different datasets; then a bilinear interpolation on the model grid was applied; and finally, a hand-made check for isolate grid points, islands and narrow passages was conducted. The three datasets used were: the Digital Bathymetric Data Base-Variable Resolution (DBDB) at 1-min resolution for the Mediterranean basin, DBDB-2 (2-min resolution) for the Atlantic box, and the very high resolution digitalized chart of Sanz et al. (1991) for the Strait of

Gibraltar. Vertical eddy viscosity and diffusivity coefficients were computed using the turbulence closure model developed by Bougeault and Lacarrere (1989) for the atmosphere and adapted for the oceanic case by Gaspar and Lefevre (1990).

Initial conditions for temperature and salinity were obtained from the Mediterranean Data Archaeology and Rescue (MEDAR) / Mediterranean Hydrological Atlas (MEDATLAS II) database (MEDAR Group, 2002). The model is forced at the surface by the atmospheric pressure, wind stress and the heat and fresh water fluxes provided by the ECMWF ERA40

reanalysis database (provided by the European Centre for Medium-Range Weather Forecasts), at a temporal resolution of six hours and a spatial resolution of about 1.125° x 1.125°, while the climatological river discharge was prescribed according to Struglia et al. (2004) for the main 68 catchments (Sannino et al., 2015). According to these authors, the Black Sea net flow through the Dardanelles was imposed by following Stanev et al. (2000).

Tides are incorporated in the model and tidal forcing includes both the tide generating potential as a body force in the

momentum equations, and the lateral boundary condition in the open Atlantic boundary imposed by the tidal velocities produced by the barotropic tidal model of Carrere and Lyard (2003) (Naranjo et al., 2014). The validation of this configuration of the model was conducted through the comparison of the modeled tides (2D) and currents (3D) with observations and previous modelling studies. In order to check the reliability of tides, a harmonic analysis (Foreman, 1977)



of the simulated sea surface height over the entire Mediterranean basin was conducted in Sannino et al. (2015) through a barotropic experiment in which only the internal and equilibrium tidal forcing were prescribed. These authors reported that the computed amplitude and the phase of the principal semidiurnal ($M_2$ and $S_2$) and diurnal ($O_1$ and $K_1$) tidal constituents showed in general a good agreement with the tide gauge values reported in the basin while a reasonable agreement was found in the Strait of Gibraltar with amplitudes differing no more than 5 cm for $M_2$ and 7 cm for $S_2$ and deviations in phase around 18° for $M_2$ and 16° for $S_2$.

To check the reliability of 3D tidal currents, we compared simulated amplitude and phase of the mean velocity vertical profiles at the main sills of ES and CS (see Fig. 1) with in-situ Acoustic Doppler Current Profiler (ADCP) observations collected in the frame of the INGRES Projects (see Sánchez-Román et al., 2008, 2009; and Sammartino et al., 2015 for details). The vertical structure of modeled mean currents in ES and CS (Figure not shown) show the two-layer character of the flow with an upper layer flowing towards the Mediterranean Sea and a lower layer flowing towards the Atlantic Ocean. They present a general good agreement with the mean profiles obtained from observations exhibiting a correlation coefficient greater than 0.90. The mean depth of the modeled interface between incoming and outgoing waters only differs in 11 m with the one computed from observations in both locations. Furthermore, we found discrepancies in amplitudes lower than 10 cm s$^{-1}$ while deviations in phase lower than 15° were observed. The reader is referred to Sannino et al. (2015) for further details about the model description and 2D validation.

The model outputs used here are 3-hourly data from two runs (with and without tidal forcing) both spanning a ten years period: from January 1958 to November 1967. The outputs were analysed at five cross-Strait sections to investigate the spatial variability of the vertical transfers between the incoming and outgoing waters during their passage through the Strait: the two sections located at the boundaries of the Strait (Gulf of Cadiz, CA, and Alboran Section, AL) and the internal sections of Espartel (ES), Camarinal Sill (CS), and Tarifa Narrows (TN; see Fig. 1b). According to results reported by Sannino et al., (2015), the model slightly overestimates the temperature and salinity of the Mediterranean basin. This fact leads to a long-term drift in the Mediterranean salinity (see Fig. 15 in Sannino et al., 2015), which suggests that the model does not reach an equilibrium state concerning the salinity of the whole basin. Thus, the 10-year mean estimates of transports reported in section 3 could be contaminated by this model drift, as they partially depend on the salinity difference between inflowing and outflowing waters; however, the vertical transfers of water, heat and salt between layers at the Strait of Gibraltar do not depend critically on the absolute value of the salinity of the basin. Moreover, because we focus on tidal to monthly frequencies, a small long-term trend is not expected to modify the basic mechanisms analysed here.

## 2.2 Computation of water, heat and salt horizontal transports

The exchanges are characterized in the framework of an inflow layer flowing eastward from the Atlantic to the Mediterranean and an outflow layer flowing westward from the Mediterranean to the Atlantic. The volume transport associated with the inflow (outflow) is computed integrating the positive (negative) velocities at a given section:



$$Qin(x,t) = \int_{South}^{North} \int_{bottom}^{\eta} u^+(x,y,z,t)dzdy,$$

(1)

$$Qout(x,t) = \int_{South}^{North} \int_{bottom}^{\eta} u^-(x,y,z,t)dzdy$$

where $u$ is the velocity component along the strait and the superscript + (-) denotes positive (negative) values. It is worth noting that this definition is different from the traditional choice of defining the layers using a given salinity surface (see e.g. García-Lafuente et al., 2000; Sannino et al., 2004) or the maximum vertical shear of the horizontal velocity (see e.g. Tsimplis and Bryden, 2000; Sánchez-Román et al., 2009) as interface, which discriminate between Atlantic and

5  Mediterranean Waters via a material (or not) surface that physically separates both water masses. Conversely, here we aim at discriminating between inflowing and outflowing waters by following a strictly velocity criterion which does not require any further definition of an interface. For each time step, we compute the amount of water that enters (exits) the Mediterranean basin regardless of its origin (Atlantic Ocean or Mediterranean Sea). We think that this approach provides a better view of the dynamical processes occurring in the Strait and will be complementary to previous descriptions based on the traditional

10  approach. In any case, the differences between the definitions will only be significant when analysing the high frequency variability, while for the subinertial variability the results are practically equivalent. The net volume transport will be the sum of the inflow and outflow components:

$$Qnet(x,t) = Qin(x,t) + Qout(x,t) = \int_{South}^{North} \int_{bottom}^{\eta} u(x,y,z,t)dzdy \qquad (2)$$

Analogously, the heat (QH) and salt (QS) fluxes associated with the inflow (outflow) are computed as:

$$QHin(x,t) = \int_{South}^{North} \int_{bottom}^{\eta} \rho C_e u^+ T^+ dzdy,$$

$$QHout(x,t) = \int_{South}^{North} \int_{bottom}^{\eta} \rho C_e u^- T^- dzdy, \qquad (3)$$

$$QSin(x,t) = \int_{South}^{North} \int_{bottom}^{\eta} u^+ S^+ dzdy,$$



$$QSout(x,t) = \int\limits_{South}^{North} \int\limits_{bottom}^{\eta} u^- S^- dz dy$$

where ρ is the density, $C_e$ the specific heat of seawater, and the superscripts + (-) in T and S indicate the model cells where the velocities are positive (negative). The heat flux will be expressed in $W/m^2$ (considering a Mediterranean Sea surface of $2.5 \cdot 10^{12}$ $m^2$) for an easy comparison with Mediterranean surface heat fluxes.

The water and salt transports together with the heat fluxes are estimated from the 3-hourly fields, which capture the tidal

variability, and also from the monthly means to investigate the low frequency variability of the exchanges. It is worth noting here that the monthly averages of 3-hourly transports and the transports computed from monthly data are not exactly the same. Because tidal currents are larger than the mean flow of the slow-flowing layers, the Atlantic inflow west of CS and the Mediterranean outflow east of CS reverse almost every tidal cycle. In those situations Qin west of CS and Qout east of CS are zero, which modifies the averaging. In practice this means that the values computed averaging high frequency transports

are higher than the transport computed from monthly averages.

### 2.3 Estimation of the water recirculation between layers

The sketch of Fig. 2 illustrates the procedure followed to estimate the vertical transfer of water between layers. It schematizes the inflowing and outflowing waters bounded by two consecutive cross-Strait sections (any of those shown in Fig. 1b), by the sea surface and the bottom.

The continuity equation establishes that the difference in the net transport between two consecutive sections must be compensated by a change in the mean free surface elevation in between the sections:

$$Qnet_{i+1} - Qnet_i = -A \frac{\partial \bar{\eta}}{\partial t} \tag{4}$$

where sub-indices $i, i+1$ denote the section and $\eta$ is the surface elevation, with the overbar denoting its average over the area (A) between sections. Decomposing the net transport into the inflowing and outflowing layers leads to:

$$Qin_{i+1} - Qin_i = -A \frac{\partial \bar{\eta}}{\partial t} + \phi \tag{5}$$

$$Qout_{i+1} - Qout_i = -\phi$$

where $\phi$ represents the vertical flux between the inflow and outflow layer (positive upwards) given by the

convergence/divergence observed at the outflow layer. Note that $\phi$ stands for the total vertical transfers between the incoming and outgoing layers and it will balance the transports between sections. Thus, it will also contribute to transient vertical excursions of the sea surface elevation between sections at tidal frequencies so it is not the flux effectively




recirculated. As a result, we must define another parameter to identify the amount of water that effectively contributes to increase the inflow or the outflow.

We define the effective recirculation flux $\xi$ as the amount of water transferred from one layer to the other changing the sign of its horizontal velocity. In other words, the amount of water of the inflow (outflow) layer that crosses section $i$ $(i+1)$, does not reach section $i+1$ $(i$ ) and comes back westward (eastward) contributing to the flow of the outflowing (inflowing) layer at section $i$ $(i+1)$. Therefore, $\xi$ will be a fraction of $\phi$. If the surface elevation was constant, the recirculation flux would be exactly equal to $\phi$. However, when there is a convergence of the outflow, for instance, the amount of water flowing upwards to the inflow layer can contribute to increase the inflow, but part of it can also contribute to elevate the free surface and return to the outflow layer afterwards. Determining the fraction of $\phi$ that corresponds to the effective recirculation flux $\xi$ is a difficult task, particularly at tidal frequencies, for which the excursions of the surface elevation can be very large. The condition we set to estimate the effective recirculation flux $\xi$ is:

$$if\ \phi > 0 \quad then \quad \xi = min(\phi, Qin_{i+1}) \tag{6}$$

$$if\ \phi < 0 \quad then \quad \xi = max(\phi, Qout_i)$$

That is, the effective recirculation flux from the inflow (outflow) layer to the outflow (inflow) layer can never be larger (in absolute terms) than the magnitude of the outflow (inflow) at that time.

## 2.4 Water mass transformation through the recirculation flux between layers

The transformation of the water masses properties in their path along the Strait can be characterized from the heat and salt fluxes between the inflowing and outflowing layers. The difference in the heat fluxes between two sections can be due to the vertical advection of temperature associated with the recirculation flux, the heat flux through the sea surface ($F_{surf}$) and the mixing between layers, which can be expressed as $\rho C_e \phi_{mix} \Delta T$, where $\phi_{mix}$ has units of water flux:

$$QHin_{i+1} - QHin_i = \rho C_e \xi \tilde{T} + \rho C_e \phi_{mix}(T_{out} - T_{in}) + F_{surf}$$
$$\tag{7}$$
$$QHout_{i+1} - QHout_i = -\rho C_e \xi \tilde{T} + \rho C_e \phi_{mix}(T_{in} - T_{out})$$

where $T_{in}$ and $T_{out}$ represent the averaged temperature of the inflow and outflow layer, respectively, and $\tilde{T}$ is equal to $T_{in}$ ($T_{out}$) when the recirculation flux $\xi$ is negative (positive). The heat advection is defined as $\rho C_e \xi \tilde{T}$ while the contribution of mixing is estimated as the residual between the heat flux difference between consecutive sections and the heat advection. In other words, it is computed as the part of the heat flux transformation between consecutive sections that cannot be explained by the advection through the recirculating fluxes. The relative importance of heat advection and mixing will be assessed in section 3.



The difference in the salt transport between two sections takes an analogous form, except that there is no salt flux through the sea surface:

$$QSin_{i+1} - QSin_i = \xi \tilde{S} + \phi_{mix}(S_{out} - S_{in})$$

$$QSout_{i+1} - QSout_i = -\xi \tilde{S} + \phi_{mix}(S_{in} - S_{out})$$

(8)

Here $S_{in}$ and $S_{out}$ represent the averaged salinity of the incoming and outgoing layer, respectively. The salt advection and salt mixing are estimated analogously to the heat terms in Eq. (7).

All the estimates shown in the following have been computed both using 3-hourly data, in order to characterize the processes occurring at high frequency, and using monthly averages, in order to characterize the low frequency processes.

**3 Results**

**3.1 Transport estimates at the boundaries of the Strait**

Figure 3 presents 3-hourly time series of the exchange flows computed at the outer limits of the Strait. The incoming positive

flow at the Gulf of Cadiz (westernmost section, blue line, upper panel) has a 10-yr mean value of $0.77 \pm 0.64$ Sv towards the Mediterranean (see Table 1). It fluctuates according to the tidal cycle showing larger values (up to 2.5 Sv) during ebb tides. Conversely, the inflow dramatically diminishes during the flood tide, when the westward barotropic tidal transport opposites the mean flow in the upper layer (Baschek et al., 2001). Tidal currents in this layer are stronger than the mean inflow, making the latter to periodically reverse (Sánchez-Román et al., 2012). As a consequence, during part of the tidal cycle the

eastward transport is zero in this area. On the other hand, the outgoing negative flow (blue line, middle panel in Fig. 3) has a 10-yr mean value of $-0.69 \pm 0.27$ Sv (negative values indicate transport towards the Atlantic Ocean). This transport also fluctuates with tides showing larger values (up to -1.8 Sv) during the flood part of the tidal cycle. This fluctuation, however, is much lower than that observed for the inflow because in this region of the Strait most of the tidal flow moves through the inflow passive layer (see e.g. Sánchez-Román et al., 2012). As a result, tidal currents are insufficient to stop the mean

outflow. The net transport (blue line, bottom panel in Fig. 3) also shows the tidal fluctuation and changes its sign in each tidal cycle. It has a 10-year mean value of $0.08 \pm 0.87$ Sv.

At the Alboran section (eastern limit of the Strait, red lines in Fig. 3), the instantaneous transports exhibit an opposite behaviour because most of the tidal flow moves through the outflow layer that acts there like the passive one. The incoming flow (red line, upper panel) only stops down during the flood part of the tidal cycle (westwards moving) of the most

energetic spring tides. This flow has a 10-yr mean value of $0.91 \pm 0.77$ Sv (Table 1) and shows maximum transports (up to 3.5 Sv during the ebb tide) larger than those of the incoming flow at the western limit of the Strait. Conversely, the outgoing flow (red line, middle panel) has a 10-yr mean value of $-0.84 \pm 0.84$ Sv reaching absolute maximum values of around -3.0 Sv during the flood tide. This transport collapses during most of the ebb tide cycles, when the tidal current opposites the mean



flow (i.e., no water escapes from the Mediterranean during this part of the tidal cycle). The net transport through the eastern limit also shows the tidal fluctuation and has a 10-year mean of 0.08 ± 1.50 Sv (Table 1). This mean value is obviously the same than at the western limit and corresponds to the basin mean value of the E-P-R budget of the model. This budget is mainly driven by the evaporative cycle of the Mediterranean and exhibits a clear seasonal signal peaking in late summer
(Soto-Navarro et al., 2010). However, the tidal variability at the eastern boundary is almost twice that in the western limit (see also the bottom panel in Fig. 3).

At subinertial frequencies the exchange is greatly reduced (Fig. 4). The mean inflow is 0.68 Sv and 0.62 Sv at the Gulf of Cadiz and Alboran sections, respectively, while the mean outflow is -0.61 and -0.55, respectively. The main mechanisms behind the subinertial variability of the flow is the variation of atmospheric pressure over the Mediterranean and, on a second
order, the changes in the Mediterranean freshwater budget and in the density difference between the exchanged waters (García-Lafuente et al., 2002). All those mechanisms are included in the model, which shows a monthly standard deviation (STD) in the inflow transport of 0.09 Sv at the Gulf of Cadiz section and of 0.13 Sv at the Alboran section (see Fig. 4 and Table 1). The subinertial variability of the outflow transport is lower (0.06 Sv at both sections).

As stated above, the difference between the mean exchanges at high and low frequencies is due to the positive correlation of
tidal currents and tidally induced vertical displacements of the interface (Sannino et al., 2004; Vargas et al., 2006; Sánchez-Román et al., 2009). The eddy-fluxes produce a tidal rectification of the flow and their contribution to the mean exchange depends on the location of the cross-Strait section (Garcia-Lafuente et al., 2000; Vargas et al., 2006; Sánchez-Román et al., 2009). As a rule of the thumb, the larger the tidal excursions of the interface, the higher the contributions of the eddy fluxes are to the mean flow. This reflects in the results, which show a contribution of the eddy fluxes to the mean transport of 0.08
± 0.02 Sv at the western limit and of 0.28 ± 0.04 Sv at the eastern limit of the Strait (Table 1).

The variability of the water transports is directly reflected in the variability of the heat exchanges (Table 1). At high frequencies, the mean heat flux at the Gulf of Cadiz section is 21.07 W/m$^2$ towards the Mediterranean and -15.99 W/m$^2$ towards the Atlantic. It results in a net heat flux of 5.09 W/m$^2$. The strong tidal variability also implies large variations of the heat flux, which shows a 3-hourly STD of 17.59, 6.67 and 23.22 W/m$^2$ for the inflow, outflow and net flow, respectively. At
subinertial frequencies the exchange is consistently reduced, with a mean inflow (outflow) of 18.83 W/m$^2$ (-13.83 W/m$^2$). The low frequency variations are also smaller than at high frequency and not only depend on the changes in the water flux, but also on the changes of the temperature of the incoming and outgoing waters. The monthly STD for the inflow, outflow and net flow are 3.32, 1.75 and 3.36 W/m$^2$, respectively. The contribution of the eddy fluxes to the mean heat exchange at the western limit is 2.06 ± 0.56 W/m$^2$. At the Alboran section the heat exchanged is larger with an averaged heat inflow of
23.38 W/m$^2$ and a heat outflow of -18.22 W/m$^2$. The net heat flux is almost the same (5.16 W/m$^2$). The heat variability at tidal frequencies in this area is larger than at the western limit with a 3-hourly STD of 18.81, 18.47 and 34.65 W/m$^2$ for the inflow, outflow and net flow, respectively. The monthly mean and STD are 16.87 ± 4.02, -11.82 ± 1.67 and 5.04 ± 3.42 W/m$^2$. As expected, the contribution of the eddy fluxes is also larger in this area (6.33 ± 0.99 W/m$^2$).



At high frequencies, the mean salt transport through the Gulf of Cadiz section is of $27.90 \times 10^6$ kg/s in the inflow layer and of $-25.97 \times 10^6$ kg/s in the outflow layer, which results in a net transport of $1.93 \times 10^6$ kg/s. The respective 3-hourly STD are 23.16, 9.82 and $31.64 \times 10^6$ kg/s. The monthly means and standard deviations are $22.23 \pm 8.58$, $-20.59 \pm 7.67$ and $1.64 \pm 3.68 \times 10^6$ kg/s, for the inflow, outflow and net flow, respectively. Notice that a net salt transport towards the Mediterranean

Sea is obtained at both tidal and low frequencies due to a slight overestimation of the model salinity in the Mediterranean basin (Sannino et al., 2015). The eddy fluxes contribution to the mean salt transport is $2.77 \pm 0.77 \times 10^6$ kg/s. consistently with what has been shown for the water and heat fluxes, at the eastern side the salt fluxes are higher and have higher variability at high frequency, while they are lower and have lower variability at low frequency (Table 1).

### 3.2 Overall transformation of the exchange along the Strait

The difference between the fluxes that cross the Gulf of Cadiz and Alboran sections provides a measure of the vertical recirculation taking place within the Strait. Fig. 5 displays the vertical transfers $\phi$ of water, estimated as the difference between the outflowing waters measured at the outer sections according to Eq. (1), and the effective recirculation flux $\xi$ computed according to Eq. (6).

The 3-hourly vertical transfer and recirculation of water exhibit large fluctuations (respective STD of 0.63 Sv and 0.38 Sv,

Table 1), shifting their sign according to the tidal cycle. Positive values are observed during ebb tides (eastward moving), then suggesting that part of the outflowing waters will be brought towards the inflow layer. Conversely, negative values are obtained during flood tides (westward moving), which implies that a fraction of the inflowing waters will be conveyed towards the outflow layer during this part of the tidal cycle. It is worth noting that only a fraction of the upward vertical transfer results in an effective recirculation flux as defined in Eq. (6); the remainder will contribute to the rising of the sea

surface elevation in between the two sections, according to Eq. (5.a). Conversely, vertical transfer and effective recirculation coincide for downward transports. This makes that the long-term averages of vertical transfer and effective recirculation flux are quite different (see Table 1): the 10-yr average of the vertical transfer is 0.14 Sv (positive meaning a net transfer from the outgoing layer to the incoming layer) while the 10-yr average of the recirculation flux is -0.07 Sv (negative meaning a net transfer from the incoming layer to the outgoing layer). It is important to highlight here that there is no inconsistency in the

discrepancies between the vertical transfer and the recirculation flux. At tidal frequencies there is a strong vertical transfer of water between layers, which mostly contributes to the rising or falling of the free surface, but which does not imply an effective recirculation of water.

At low frequency the picture changes and the net vertical transfer matches the net recirculation of water (-0.07 Sv, Table 1) which means that a fraction of the inflowing water that crosses the western boundary recirculates towards the outflow layer

and thus does not reach the Mediterranean basin. Note that, as expected, this mean value also coincides with the recirculation flow obtained at tidal frequency, since at low frequencies the sea level changes are very small. The monthly variability of the vertical transfer/recirculation flux has a STD of 0.05 Sv and only during 6% of the time changes its sign and becomes positive (Fig. 5, lower panel). The dominant positive sign of the vertical transfer at high frequency must be therefore



associated with the tidal forcing exerted over the mean flow, which promotes large vertical excursions of the interface along the tidal cycle that do not necessarily involve recirculation fluxes between layers. It seems clear that the vertical recirculation obtained above will have an impact on the hydrological properties of the exchange flows throughout the Strait, since it implies vertical transfers of heat and salt between layers. On average, the heat flux and salt transport advected along the Strait towards the outflow layer at high frequency are -1.70 W/m$^2$ and -2.00×10$^6$kg/s, respectively. This represents around 10% of the inflowing and outflowing heat and salt fluxes. The 3-hourly variability of heat (salt) recirculation shows a STD of 5.38 W/m$^2$ (15.51×10$^6$kg/s). At low frequency, the mean advection of heat and salt are -2.01 W/m$^2$ and -1.86×10$^6$kg/s, with a STD of 1.12 W/m$^2$ and 1.84×10$^6$kg/s, respectively.

On the other hand, from Eq. (7) we obtained that the heat and salt transferred between the inflow and the outflow due to mixing presents respectively a 10-year mean value of -0.03± 1.07 W/m$^2$ and -0.02 ± 1.73x10$^6$kg/s (see Table 1). Therefore, a major conclusion that can be derived from these results is that the dominant mechanism driving the heat and salt transfer between layers is the water advection, while mixing has a much smaller contribution. In order to check it, from Eq. (7,8) we can obtain a first estimate of the heat transferred by water advection ($\rho C_e \xi \tilde{T}$): using the averaged recirculation flux at high/low frequency ($\xi = -0.07$ Sv, Table 1) and the mean value of the inflow layer temperature in the Gulf of Cadiz ($\tilde{T}$=16.84, Table 2) we get a value of -1.94 W/m$^2$, very close to the total heat transfers of -1.70W/m$^2$ and -2.01W/m$^2$ at high and low frequencies, respectively (Table 1). Therefore, the mixing between the layers is expected to play a secondary role in the vertical exchange of heat and salt along the Strait.

As a result of the advection of heat and salt, the averaged temperature and salinity of each layer is transformed along the Strait. The incoming waters computed at the Gulf of Cadiz section, composed mostly of SAW and NACW, have a 10-yr mean temperature (salinity) of 16.84°C (36.28 psu) with a monthly standard deviation of 1.12°C (0.04 psu) mainly induced by a combination of the seasonal cycle and the recirculation of water from the outflow layer (Table 2). Garcia-Lafuente et al. (2011) suggested that NACW, colder and fresher than SAW, does not overpass the threshold of Camarinal because it is swept along the backwards outflow layer in the Tangier Basin (see Fig. 1). Temperature-salinity diagrams (not shown) conducted at the different cross-sections of the Strait analysed here confirm this fact. This by itself would imply that the water reaching the Mediterranean, composed mostly of SAW, would be warmer and slightly saltier than the incoming waters. Our results do show saltier (37.15±0.04 psu) waters in the inflow layer of the Alboran Section, but they have a 10-yr mean temperature of 15.68 ±1.25°C, that is, 1.16°C colder than at the western Strait. This implies the addition to the incoming flow of a fraction of the colder and saltier outflowing waters due to the tidally-induced recirculation fluxes between layers. The outgoing waters, composed mostly of MOW (a mixture of LIW, WIW, TDW and WMDW), have at the eastern boundary of the Strait a 10-yr mean temperature and salinity of 14.06 ± 0.44°C and 37.87 ± 0.10 psu, respectively. This flow becomes warmer and fresher as it travels to the Atlantic Ocean due to the aforementioned tidally-induced recirculation of NACW west of CS. As a consequence, the outflowing waters exit the Strait with a 10-yr mean temperature and salinity of 15.00 ± 0.28°C and 36.99 ± 0.05 psu, respectively. These values are summarized in Table 2.



The transformation of the properties of the inflow and the outflow along their passage through the Strait is not constant in time. As an example, the monthly time series of the temperature at both sides of the Strait is shown in Fig. 6 (analogous results are found for salinity, figure not shown). A clear seasonal cycle (STD = 1. 12°C) modulates the inflow at both sections, as it is the decrease along the Strait of the temperature of the incoming layer (-1.16°C, with a relatively low

monthly STD of 0.14°C). Concerning the outgoing layer, the temperature change is on average smaller (it increases 0.94°C along its westward path), but the variability is larger (monthly STD of 0.34°C). During the periods when the vertical recirculation between layers is small or positive (see Fig. 5, lower panel), the temperature of the outflow at both sides of the Strait is almost the same.

As a consequence of the water transformation along the Strait, the difference between the inflow and outflow properties is

neither constant in time or in space (Table 3). The difference in the temperature of both layers at the Gulf of Cadiz section is 1.83 ± 0.78 °C while it is slightly lower (1.61 ± 0.52°C) at the Alboran section. There is a marked seasonal cycle for the difference at both sides peaking in August and with a minimum in February. The difference in the salinity of both layers is - 0.71 ± 0.06 psu and -0.72 ± 0.11 psu, respectively, thus being slightly larger at the eastern side. In this case there is no clear seasonal cycle in the salinity difference. As a result, the density difference between layers is -1.18 ± 0.21 kg/m$^3$ in the Gulf

of Cadiz section and -1.09 ± 0.18 kg/m$^3$ in the Alboran section. During winter the density difference between layers is the lowest and is almost the same at both sides of the Strait (∼0.8 kg/m$^3$). In summer the density difference increases, being ∼1.45 kg/m$^3$ and ∼1.3kg/m$^3$ at the Gulf of Cadiz and Alboran sections, respectively (figure not shown). The changes in the density difference have a direct impact on the low frequency exchange flows, with larger density differences implying larger exchanges. The correlation between the density difference and the inflow is significant at both sides (0.59 and 0.65 at the

Gulf of Cadiz and Alboran sections, respectively).

### 3.3 Water recirculation along the Strait

In order to gain insight in the flux transformation occurring in the Strait, the horizontal transports of volume, heat and salt were also computed at the internal sections (ES, CS and TN) at both high and low frequencies (see Tables 4 and 5). The most remarkable fact is that exchanges are very variable along the Strait. For the tidal induced variability (Table 4 and Fig.

7) and from west to east the mean exchange increases from the Gulf of Cadiz to Espartel and then decreases to a minimum in CS. After CS the exchange increases again reaching a maximum at the easternmost section. This behavior is in good agreement with previous studies (Garcia-Lafuente et al., 2000; Garcia-Lafuente et al., 2013) that already showed that tidal currents are stronger in the eastern part of the Strait. However, it is worth noting that the exchange at Espartel is larger than at the neighboring sections. This variability in the exchanges implies strong recirculation fluxes of water in between the

sections. In particular, between CA and ES the averaged recirculation flux is -0.06 Sv, between ES and CS it is -0.11 Sv, between CS and TN it is +0.11 Sv and between TN and AL +0.01 Sv. East of CS the variability of the vertical recirculation is large (3-hourly STD is ∼0.25 Sv everywhere). The pattern of heat and salt advection between layers is consistent with the





water recirculation fluxes: the highest vertical transfers of properties are between the sections of ES and CS and, in second place, between CS and TN.

Relating the high frequency recirculation fluxes with the phase of the tidal flow provides also an interesting picture. During flood tide (westward barotropic tidal transport, Fig. 7b) the transport in the outflow layer is the highest (-0.89 Sv at CA and -

1.57 Sv at AL) and the net vertical recirculation is positive in between all sections (+0.40 Sv in total). The recirculation flux is especially strong between CS and TN (0.37 Sv) and between ES and CS (0.15 Sv). During ebb tide (eastward barotropic tidal transport, Fig. 7c), the transport in the inflow layer is higher (1.29 Sv at CA and 1.48 Sv at AL) and the net vertical recirculation is negative in between all sections (-0.38 Sv in total). In this case, the largest recirculation flux is between ES and CS (-0.23 Sv) and between CA and ES (-0.13 Sv), while it is almost negligible east of CS.

At low frequency the picture is quite similar (Table 5 and Fig. 8a). There is a steady decrease of transports from both the Gulf of Cadiz and Alboran sections to the main sill of Camarinal, where minimum values of the exchange flows are obtained (0.48 Sv and -0.41 Sv for the inflow and outflow, respectively). Therefore, west of CS there is a net vertical recirculation flux towards the outflow layer while east of CS the net vertical transfer is positive and contributes to increase the water exchange. This behavior was reported by Garcia-Lafuente et al. (2000) east of CS for both the inflow and outflow and, more

recently, Garcia-Lafuente et al. (2011) showed the same pattern between ES and CS. The largest recirculation fluxes are found between ES and CS (-0.14 Sv) and between CS and TN (+0.13 Sv) but on average the net vertical recirculation of water along the Strait is negative (-0.07 Sv). The time variability of the recirculation flux is similar everywhere (monthly STD = 0.02 Sv) and except between TN and AL never implies a change in its sign.

## 4 Discussion

**4.1 Forcing of the recirculation fluxes between incoming and outgoing waters**

In order to investigate the mechanism behind the reported recirculation fluxes between layers we first computed the correlation between the high frequency vertical recirculation and the inflow, the outflow and the net transport. The results suggest that the recirculation fluxes are driven by the outflow variability, since the highest correlations are found with the outflow in between all sections. In particular, the correlation of the outflow with the recirculation flux is -0.64, -0.66 and -

0.55 between CA-ES, ES-CS and CS-TN sections. This means that when the outflow increases its magnitude (more negative), the vertical recirculation towards the inflow layer is increased. Nevertheless, it is also worth noting that between CS-TN, where the mean vertical recirculation exhibits maximum positive values (+0.11 Sv, Table 4) the recirculation flux is positive most of the time. In the TN-AL region, the correlation between the vertical recirculation and the exchanges is very low (<0.2). However, there is a high correlation (0.70) between the recirculation flux and the tendency of the outflow

($dQ_{out}/dt$). This would mean that in this region it is not the intensity of the outflow which drives the vertical recirculation between layers, but the changes in the intensity. This would explain why the mean recirculation flow during ebb and flood tide is almost zero in this region (see Fig. 7) while the variability is much higher (0.23 Sv): as the tendency of the outflow is





out of phase with the flow, when averaging for positive/negative phases of the tidal flow we include positive and negative vertical recirculation fluxes between layers, leading to an almost zero averaged recirculation flux.

The results presented so far are consistent with the picture presented by Sánchez-Román et al. (2012) based on current measurements at different sites along the Strait. These authors reported that strong internal divergences taking place between

different cross-Strait sections are responsible for vertical excursions of the interface as large as one hundred meters and also for the transfer of tidal signals between the incoming and outgoing layers and vice-versa. They suggested that during flood tide (westward tidal transport), the flow is hydraulically controlled at CS and only a limited volume of the water flowing westward through AL is able to surpass the sill. They stated that the rest of the westward flow remains trapped between the two sections; here we have identified a positive recirculation flux towards the inflow layer. At the ES section, west of CS,

the westward moving flow is also hydraulically controlled and the reduced fraction of the tidal transport crossing CS is not even able to overflow ES. Consequently, the water accumulates in the Tangier basin, where a positive vertical recirculation of water has again been obtained. Conversely, during ebb tides (eastward tidal transport) the hydraulic control in CS floods by the end of the flood tide, when the tidal current weakens. This fact allows the outflowing waters accumulated in the Tangier Basin to flow eastward, thus reversing the flow in this layer at CS. On the other hand, the westward moving water

accumulated between CS and the Alboran section during the previous flood tide evacuates toward the Mediterranean through the latter. This process is favoured by the hydraulic control in Tarifa narrows (Fig. 1), which prevents the large volume of inflowing water moving to the east during the ebb tide to cross the control section. As this water accumulates between the main sill of Camarinal and the Alboran section, it pushes down the interface and forces the outflowing water below to flow back toward the Mediterranean Sea, which in our case is interpreted as a negative recirculation flux towards the outflow

layer. It is worth noting that the behaviour between ebb and flood tide is not symmetrical (Fig. 7): the magnitude of the positive vertical recirculation between layers is larger than the magnitude of the negative recirculation fluxes between CA and ES and between CS and TN, while it is the opposite between ES and CS.

At low frequency and east of CS, the uppermost fraction of the outflowing waters is systematically recirculated towards the Mediterranean. The averaged recirculation flow at high and low frequencies (Fig. 7a and 8a respectively) are in good

agreement, which suggests that the recirculation of water at low frequency is in fact the result of the residual recirculation processes of the high frequency variability. West of CS, the lower part of the inflow recirculates towards the Atlantic Ocean before reaching the CS (Fig. 8a). This is in good agreement with the observations of García-Lafuente et al. (2011) who have reported that NACW do not overpass the Camarinal Sill.

In order to give more light to the mechanism behind the recirculation processed at low frequency, we also computed the

recirculation fluxes from a simulation run without tidal forcing (Fig. 8b and Table 6).

The obtained values are very similar to those of the tidal run between CA and ES and between TN and AL, while they are clearly different between ES and CS and between CS and TN. In those regions, the recirculation of water of the non-tidal run



are much smaller (-0.05 Sv in ES-CS and 0.00 Sv in CS-TN, while in the tidal run those values were -0.14 Sv and +0.13 Sv, respectively). In the absence of tidal forcing, a possible explanation for the recirculation of water from one layer to other is that the vertical transfers are mainly induced by the dragging of the passive layer (outflowing layer to the east, inflowing layer to the west) by the active one (inflowing layer to the east, outflowing layer to the west). That is, the mechanical drag

would explain most of the recirculation processes at low frequency in the CA-ES and TN-AL regions and about a third of the recirculation of water in the ES-CS region. The tidal forcing would be responsible for the rest of the recirculation taking place in the ES-CS region and for all the recirculation processes in the CS-TN region.

### 4.2 Consequences of the water mass transformations along the Strait for the Mediterranean

The inclusion of tidal forcing in the model simulation implies a strong transformation of the exchanged water masses along

their path across the Strait. As quantified before, the inflow layer flow is 1.16°C colder and 0.86 salinity units saltier when it enters the western Alboran Sea than at origin, then suggesting the addition to the inflow of a fraction of the colder and saltier outflowing waters due to the vertical transfers between layers. Additionally, the outgoing waters that exit the Strait at its westernmost part (Gulf of Cadiz section) is 0.94°C warmer and 0.87 salinity units fresher than the one observed at the Alboran section. When no tidal forcing is included in the simulation, the water mass transformations are weaker (Table 2).

The temperature of the inflow layer decreases 0.66°C and becomes 0.47 psu saltier. Similarly, the transformation of the outflowing water is also milder than in the run including the tidal forcing. It is important to notice that the non-tidal simulation still includes the high frequency variability due to changes in the atmospheric pressure over the Mediterranean and it has enough spatial resolution to properly solve the interactions between the flow and the topography. In lower resolution models and/or runs without atmospheric pressure forcing (e.g. in climate runs with typical resolution of 1/12°,

Soto-Navarro et al., 2015), the transformation of water properties along the Strait are even weaker.

An adequate modeling of the water transformation along the Strait is crucial for the modeling of the Mediterranean long term evolution. The tidally-induced cooler and saltier AW crossing the Alboran section reaches almost the entire Mediterranean filling the upper 250 m in the western part of the western basin and deeper layers further east. Harzallah et al. (2014) investigated the impact of tidal oscillations on the thermohaline circulation of the Mediterranean Sea from two parallel

multi-decadal numerical experiments conducted with and without tides. They reported a Mediterranean 0.08°C cooler and 0.012 salinity units saltier after the simulation period (spanning from 1957 to 2007) for the tidal run. This would be a direct consequence of the recirculation of heat and salt taking place along the Strait, which acts as an inhibiting mechanism for the renewal of Mediterranean waters.

Harzallah et al. (2014) stated that the impact of considering tides is particularly important in the upper and intermediate

layers of the Mediterranean basin, leading to more homogenized waters. The Atlantic Water of these layers becomes saltier and denser while progressing into the basin and most of this flow returns to the Atlantic Ocean as LIW, formed during winter convection in the Levantine sub-basin. Another part is transformed into Eastern Mediterranean Deep Water (EMDW) in the



Adriatic and the Aegean sub-basins and into WMDW in the Gulf of Lions. All processes of deep-water formation involve LIW to a lesser or greater extent, which makes that all water masses are closely related and that any significant modification to one of them may propagate its effect to the others (Criado-Aldeanueva et al. 2012). Thus, changes in the buoyancy of the inflow entering the Alboran Sea as a result of an enhancement/decrease in the vertical transfer of heat and salt between

layers at Gibraltar will have an impact for instance on the aforementioned deep convection processes that lead to the winter formation of WMDW due to the preconditioned buoyancy of the surface Atlantic waters that reach the Gulf of Lions coming from Gibraltar. Naranjo et al. (2014) have shown that a colder and saltier (less buoyant) inflow through Gibraltar favors the production of WMDW; warmer and fresher Atlantic waters have the opposite effect. Moreover, results showed in Sannino et al. (2015) suggest that tides induce also changes in the intermediate circulation of the Tyrrhenian Sea bringing to a better

representation of local structures and a reinforcement of the Mediterranean thermohaline cell. They have also shown that LIW dispersal paths in the eastern basin are also affected by tides.

### 4.3 Simple description of the changes along the Strait

Climate simulations are usually run without tidal forcing and with a too coarse spatial resolution as to represent explicitly the modification of the exchange flows at Gibraltar. This means that the water transformations described above have to be

represented in some way. Usually this is solved by arbitrarily increasing the vertical mixing at the Strait. In this section we intend to provide a better way of representing the transformation of properties that occur at the Strait, namely using simple relationships between the characteristics of the water masses before entering the Strait and the net flow. These relationships could be used as a proxy for the fine tuning of the parameterizations used by coarse resolution models. The relationships will be developed for low-frequency exchange flows (i.e. for those simulated in climate runs).

To do this we can take profit of the fact that the water recirculation flux depends on the intensity of the exchange, especially of the outflow. However, using a parameter (the outflow) that is affected itself by the vertical recirculation and that is currently difficult to monitor in the real ocean due to technical and operational limitations (and therefore can hardly be validated) may not be the best option. An alternative is to use the net flux, which can be estimated from the water budget closure of the Mediterranean for the last decades. It involves presently better monitored parameters such as evaporation,

precipitation and river run-off (e.g. Soto-Navarro et al., 2010; Jordà et al., 2017) which have smaller uncertainties than the ones inferred from the current monitoring of the Strait. Namely we found that the variability of the net monthly water recirculation flux at both sides of the Strait can be inferred from the net water flux through the regression equations:

$$\xi_{rec}^{Cadiz} = 0.13 \times Q_{net} - 0.18$$

$$\xi_{rec}^{Alboran} = 0.32 \times Q_{net} + 0.12 \tag{9}$$

$$\xi_{rec} = \xi_{rec}^{Cadiz} + \xi_{rec}^{Alboran}$$



where the net transport $Q_{net}$ and the reconstructed recirculation fluxes $\xi_{rec}^{Cadiz}, \xi_{rec}^{Alboran}$ are expressed in Sv. The reconstructed recirculation flux at the Gulf of Cadiz (Alboran) section shows a correlation with the actual recirculation flux of 0.86 (0.87) and a root mean square (rms) error of 0.01 (0.02) Sv. The reconstructed recirculation flux at the Gulf of Cadiz section is always negative (at monthly scales) while it is always positive at the Alboran section. This allows estimating what

should be the inflow/outflow transport at the Alboran section as a function of the transports at the Gulf of Cadiz section:

$$Q_{in\_rec}^{Alboran} = Q_{in}^{Cadiz} + \xi_{rec}^{Cadiz} + \xi_{rec}^{Alboran} = Q_{in}^{Cadiz} + 0.45 \times Q_{net} - 0.06$$

$$Q_{out\_rec}^{Alboran} = Q_{out}^{Cadiz} - \xi_{rec}^{Cadiz} - \xi_{rec}^{Alboran} = Q_{out}^{Cadiz} - 0.45 \times Q_{net} + 0.06$$

(10)

Next we can estimate the modification of the averaged properties in each layer by considering that part of the water in the inflow layer is actually water recirculated from the outflow layer (see sketch in Fig. 8a). In this way, the new averaged temperature (and analogously the salinity) at both exits of the Strait can be estimated as:

$$T_{in\_rec}^{Alboran} = \frac{T_{in}^{Cadiz} \, Qin^{Cadiz} + T_{in}^{Cadiz} \xi_{rec}^{Cadiz} + T_{out}^{Alboran} \xi_{rec}^{Alboran}}{Qin^{Cadiz} + \xi_{rec}^{Cadiz} + \xi_{rec}^{Alboran}}$$

$$T_{out}^{Cadiz} = \frac{T_{out}^{Alboran} Q_{out\_rec}^{Alboran} + T_{in}^{Cadiz} \xi_{rec}^{Cadiz} + T_{out}^{Alboran} \xi_{rec}^{Alboran}}{Q_{out\_rec}^{Alboran} + \xi_{rec}^{Cadiz} + \xi_{rec}^{Alboran}}$$

(11)

The next step is using Eq. (9) and Eq. (10) to estimate the heat flux and salt transport at the Alboran section. This is

equivalent to use Eq. (7) and Eq. (8) neglecting the mixing, which we do because we have no simple estimate for the mixing ($\phi_{mix}$) and because it was shown to be a small contribution compared to the recirculation fluxes. The heat flux and salt transport entering in the Mediterranean can then be expressed as:

$$RHF_{in}^{Alboran} = \rho C_e A_{med}^{-1} Q_{in\_rec}^{Alboran} T_{in\_rec}^{Alboran}$$

$$RSF_{in}^{Alboran} = Q_{in\_rec}^{Alboran} S_{in\_rec}^{Alboran}$$

(12)

where RHF and RSF stand for the reconstructed heat flux and salt transport and have units of W/m$^2$ and $10^6$kg/s, respectively. The reconstructed heat flux and salt transport are in good agreement with the actual values provided by the

numerical model (Fig. 9).

For the heat inflow through the Alboran section, the correlation, rms error and bias of the reconstructed heat flux (blue dots in Fig. 9, upper panel) are 0.99, 0.93 W/m$^2$ and -0.80 W/m$^2$, respectively. These are much better skills than if the values of the inflow through the Gulf of Cadiz were used (i.e. equivalent to consider that no transformation happened along the path through the Strait, as happens in the coarse resolution models). In that case, the correlation, rms error and bias would be

0.96, 1.96 W/m$^2$ and -1.86 W/m$^2$, respectively (differences between red and blue dots in Fig. 9, upper panel). The skills for





the salt transport are similar (Fig. 9, lower panel): the correlation, rms error and bias of the reconstructed salt transport are 0.98, $1.62{\times}10^6$ kg/s and $-1.27{\times}10^6$ kg/s, respectively, while they would be 0.96, $2.80{\times}10^6$kg/s and $-2.11{\times}10^6$kg/s if no transformation along the Strait was assumed.

The simple relationships derived above allow the representation of water, heat and salt vertical transports based on quantities that, in principle, coarse resolution models can properly simulate (water properties before entering the Strait and the net water transport, which is controlled by the freshwater fluxes averaged over the Mediterranean basin). Therefore, they can be used as a proxy for the fine tuning of mixing parameterizations in the Strait. However, it has to be noted that increasing the vertical mixing at the Strait does not modify the water transport in each layer, which also has an influence on the evolution of the waters in the Alboran Sea. A possible alternative for a better modelling of the Strait of Gibraltar exchanges in a coarse resolution model could be to implement a buffer zone around the Strait where the temperature, salinity and along-Strait velocities ($v$) are modified gradually from the western boundary to the eastern boundary:

$$\varphi_{new}(x,y,z,t) = \alpha(x)\varphi^{Cadiz}(y,z,t) + (1-\alpha(x))\,\varphi_{new}^{Alboran}(y,z,t) \tag{13}$$

where $\varphi$ refers to T,S and v. That is, the properties in the buffer zone would be adjusted as a linear combination of the velocities prescribed at the boundaries of the buffer zone (here denoted by super-indices *Gulf of Cadiz* and *Alboran*). The weight $\alpha(x)$ could be simply defined as a function of the distance to the eastern boundary (e.g. $\alpha(x) = \frac{x_{Alboran}-x}{x_{Alboran}-x_{Gulf\,of\,Cadiz}}$).

The new temperatures and salinities at the Alboran section would be modified in such a way that the averaged value for the inflow would be defined by Eq. (9). That is:

$$T_{new}^{Alboran}(y,z,t)\big|_{in} = T^{Alboran}(y,z,t)\big|_{in} + \left(T_{in_{rec}}^{Alboran}(t) - T_{in}^{Alboran}(t)\right) \tag{14}$$

$$T_{new}^{Alboran}(y,z,t)\big|_{out} = T^{Alboran}(y,z,t)\big|_{out}$$

Where subscript *in* (*out*) indicate the model cells where the velocities are positive (negative). The procedure would be analogous for the salinity. The new velocities at the eastern boundary of the buffer zone will have to be modified differently depending if they are positive or negative:

$$v_{new}^{Alboran}(y,z,t)\big|_{in} = v^{Alboran}(y,z,t)\big|_{in} \cdot \left(1 + \frac{\xi_{rec}}{\int v_{Alboran}(y,z,t)\big|_{in}dydz}\right)$$

$$v_{new}^{Alboran}(y,z,t)\big|_{out} = v^{Alboran}(y,z,t)\big|_{out} \cdot \left(1 - \frac{\xi_{rec}}{\int v_{Alboran}(y,z,t)\big|_{out}dydz}\right) \tag{15}$$

where $\xi_{rec}$ is computed using Eq. (9) and the last term in each equation represents the fraction between the recirculated water flux and the inflow/outflow.



This approach ensures the conservation of the net flux of water through the Strait and therefore will not produce inconsistencies in the model run. Nevertheless, dedicated experiments with a numerical model would be required to test the proposed approach and identify eventual numerical problems that could appear in its practical implementation. The test should also be useful to determine the impact of the proposed modifications in the temperature and salinity of the inflow on

the Mediterranean evolution and to compare our results with previous experiments (e.g. with Naranjo et al., 2014; Sannino et al., 2015).

## 5 Conclusions

The transformation of the water exchanges through the Strait of Gibraltar has been investigated using a numerical model of the Mediterranean Sea. Distinctive features of the model simulation are its very high spatial resolution around the Strait and

the inclusion of tidal and atmospheric pressure forcing, in addition to air-sea fluxes. In order to account for the total amount of water than enters/escapes to/from the Mediterranean basin, incoming and outgoing waters were discriminated according to a velocity approach instead of the classical criterion based on a given salinity surface, which discriminates between Atlantic and Mediterranean Waters. The model results show a complex pattern for the exchange, with large vertical transfers of water between the incoming and outgoing flows, part of which translate in an effective recirculation of water between layers. The

10-year average of the net recirculation flux along the Strait is -0.07 Sv at both tidal and subinertial frequencies, but with a large time variability (3-hourly STD = 0.38 Sv; monthly STD = 0.05 Sv).

The physical mechanism behind the obtained recirculation pattern seems to be the hydraulic control acting at the Espartel section, Camarinal Sill and Tarifa Narrows, which limits the amount of water than can cross the sections and forces vertical recirculation. At subinertial frequencies and far from the Camarinal Sill (between the Gulf of Cadiz-Espartel sections and

between the Tarifa Narrows-Alboran sections), the vertical transfers are also induced by the dragging of the passive layer (outflowing layer to the east, inflowing layer to the west) by the active one (inflowing layer to the east, outflowing layer to the west).

Traditionally, the mechanism behind heat and salt vertical transfers has been ascribed to turbulent mixing enhanced by the tidal forcing. This is actually what the coarse resolution models assume. However, we have shown that it is the recirculation

of water between layers, and not the mixing, what dominates the vertical transfer of heat and salt. Unfortunately this vertical advection of water properties and the subsequent modification of heat and salt fluxes cannot be explicitly simulated by numerical models with a spatial resolution coarser than 500 m and/or by models that do not include tides (Sannino et al., 2015). Thus, we have proposed a simple proxy for the temperature and salinity changes that occur at the Strait. This relationship could be used to fine tune the mixing parameterizations that coarse resolution models use at the Strait of

Gibraltar. Additionally we have also proposed a relaxation scheme as a way to improve the representation of the exchanges and water property transformation in coarse resolution models.



The recirculation of water between the incoming and outgoing layers implies an advection of heat and salt and thus modifies the properties of the water inflowing in the Mediterranean. This process is crucial for a proper modelling of the Mediterranean Sea at climate scales. Not considering any recirculation process between layers at Gibraltar will overestimate the heat and salt exchanges and result in an overestimation of the Mediterranean thermohaline circulation.

**Author contribution**

G. Sannino developed the model code and performed the simulations. A. Sánchez-Román processed and analysed the model outputs; and prepared the manuscript with contributions from all co-authors. G. Jordà developed section 4.3 aimed at helping coarse resolution models. D. Gomis contributed to the results and discussion sections.

**Competing interests**

The authors declare that they have no conflict of interest.

**Acknowledgements**

This work has been carried out in the framework of the projects VANIMEDAT-2 (CTM2009-10163-C02-01, funded by the Spanish Marine Science and Technology Program and the E-Plan of the Spanish Government) and CLIFISH (CTM2015-66400-C3-2-R, funded by the Spanish Ministry of Economy). A. Sánchez-Román acknowledges a Juan de la Cierva contract

(JCI-2011-10196) funded by the Spanish Ministry of Economy and Competitiveness. G. Jordà acknowledges a Ramón y Cajal contract (RYC-2013-14714) funded by the Spanish Ministry of Economy and the Regional Government of the Balearic Islands; he also acknowledges a post-doctoral grant funded by the Regional Government of the Balearic Islands and the European Social Fund. We acknowledge PRACE for awarding us access to resource FERMI based in Italy at CINECA. The in-situ measurements used to validate the numerical model have been collected during the INGRES Projects, INGRES

(REN2003_01608), INGRES2 (CTM2006_02326/MAR), and INGRES3 (CTM2010_21229-C02-01/MAR), and Special Action CTM2009-05810-E/MAR, funded by the Spanish Government.



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



**Table 1. Mean values and standard deviations of the horizontal transports of water, heat and salt at the outermost cross-strait sections (Gulf of Cadiz and Alboran). The estimated vertical transfer, recirculation fluxes and mixing (see text for details) are also indicated. The results are shown for the 3-hourly data and the monthly averages.**

| FLUXES | | Water (Sv) | | Heat (W·m⁻²) | | Salt (10⁶ kg·s⁻¹) | |
|---|---|---|---|---|---|---|---|
| | | G. Cadiz | Alboran | G. Cadiz | Alboran | G. Cadiz | Alboran |
| 3 - hourly | IN | 0.77± 0.64 | 0.91± 0.77 | 21.07 ± 17.59 | 23.38 ± 18.81 | 27.90 ± 23.16 | 33.89 ± 29.06 |
| | OUT | -0.69 ± 0.27 | -0.84 ± 0.84 | -15.99 ± 6.67 | -18.22 ± 18.47 | -25.97 ± 9.82 | -31.96 ± 32.38 |
| | NET | 0.08 ±0.87 | 0.08± 1.50 | 5.09 ± 23.22 | 5.16 ± 34.65 | 1.93 ± 31.64 | 1.93 ± 57.05 |
| | Vertical transfer | 0.14 ± 0.63 | | 2.23 ± 13.16 | | 5.99 ± 24.53 | |
| | Recirculation flow | -0.07 ± 0.38 | | -1.70 ± 5.38 | | -2.00 ± 15.51 | |
| Monthly | IN | 0.68± 0.09 | 0.62± 0.13 | 18.83 ± 3.32 | 16.87 ± 4.02 | 22.23 ± 8.58 | 20.31 ± 8.56 |
| | OUT | -0.61 ± 0.06 | -0.55 ± 0.06 | -13.83 ± 1.75 | -11.82 ± 1.67 | -20.59 ± 7.67 | -18.73 ± 7.15 |
| | NET | 0.07± 0.11 | 0.07± 0.11 | 5.00 ± 3.36 | 5.04 ± 3.42 | 1.64±3.68 | 1.58±3.68 |
| | Vertical transfer / Recirculation | -0.07 ± 0.05 | | -2.01 ± 1.12 | | -1.86 ± 1.84 | |
| Mixing | | -- | | -0.03 ± 1.07 | | -0.02 ± 1.73 | |
| Eddy Fluxes | | 0.08± 0.02 | 0.28± 0.04 | 2.06±0.56 | 6.33± 0.99 | 2.77±0.77 | 10.88 ± 1.50 |





**Table 2. Mean values and standard deviation of the monthly temperature (°C) and salinity (psu) values for the inflow and outflow at the outermost cross-strait sections (Gulf of Cadiz and Alboran). The results are shown for the run including tidal forcing and the run without the tidal forcing.**

| | TIDAL RUN | | | NON-TIDAL RUN | | |
|---|---|---|---|---|---|---|
| | Gulf Cadiz | Alboran | Difference Alboran – G. Cadiz | Gulf Cadiz | Alboran | Difference Alboran – G. Cadiz |
| Temperature IN (ºC) | 16.84±1.12 | 15.68±1.25 | **-1.16 ± 0.14** | 16.87±1.11 | 16.21± 1.08 | **-0.66 ± 0.33** |
| Temperature OUT (ºC) | 15.00±0.28 | 14.06±0.44 | **-0.94 ± 0.34** | 14.28±0.18 | 13.22± 0.24 | **-1.05 ± 0.07** |
| Salinity IN (psu) | 36.28±0.04 | 37.15±0.04 | **0.86 ± 0.06** | 36.26±0.04 | 36.74± 0.12 | **0.47±0.11** |
| Salinity OUT (psu) | 36.99±0.05 | 37.87±0.10 | **0.87 ± 0.07** | 37.70±0.09 | 38.38± 0.06 | **0.68 ± 0.07** |





**Table 3. Difference in the properties of the inflow and outflow at the boundaries of the Strait.**

| | Gulf Cadiz | Alboran |
|---|---|---|
| **T Inflow – T outflow (ºC)** | 1.83 ± 0.78 | 1.61 ± 0.52 |
| **S Inflow – S outflow (psu)** | -0.71 ± 0.06 | -0.72 ± 0.11 |
| **ρ Inflow – ρ outflow (kg/m³)** | -1.18 ± 0.21 | -1.09 ± 0.18 |



**Table 4. Mean values and standard deviations of the horizontal transports of water, heat and salt at each cross-strait section. The estimated vertical transfer and recirculation fluxes between sections are also indicated. The results are shown for the 3-hourly data.**

| | | G. Cadiz | Espartel | Camarinal Sill | Tarifa Narrows | Alboran |
|---|---|---|---|---|---|---|
| **Water Transport (Sv)** | **IN** | 0.77 ± 0.64 | 0.82 ± 0.82 | 0.77 ± 0.81 | 0.89 ± 0.75 | 0.91 ± 0.77 |
| | **OUT** | -0.69± 0.27 | -0.75 ± 0.50 | -0.69 ± 0.69 | -0.82 ± 0.82 | -0.84± 0.84 |
| | **NET** | 0.08 ± 0.87 | 0.08 ± 1.25 | 0.08 ± 1.41 | 0.08 ± 1.46 | 0.08 ± 1.50 |
| | **Vertical transfer** | 0.05 ± 0.26 | | -0.05 ± 0.23 | 0.12 ± 0.25 | 0.02 ± 0.23 |
| | **Recirculation flow** | -0.06 ± 0.09 | | -0.11 ± 0.17 | 0.11 ± 0.24 | 0.01 ± 0.23 |
| **Heat Flux (W/m²)** | **IN** | 21.07 ± 17.59 | 22.54 ± 22.23 | 21.07 ± 21.67 | 22.91 ± 18.62 | 23.38 ± 18.81 |
| | **OUT** | -15.99 ± 6.67 | -17.38 ± 12.46 | -15.84 ± 16.14 | -17.81 ± 17.83 | -18.22 ± 18.47 |
| | **NET** | 5.09 ± 23.22 | 5.16 ± 32.72 | 5.24 ± 35.70 | 5.10 ± 33.98 | 5.16 ± 34.65 |
| | **Vertical transfer** | 1.40 ± 6.37 | | -1.55 ± 4.75 | 1.82 ± 12.27 | 0.41 ± 4.95 |
| | **Recirculation flow** | -1.45 ± 1.29 | | -2.56 ± 2.27 | 1.96 ± 3.04 | 0.35 ± 2.99 |
| **Salt Transport (10⁶ kg·s⁻¹)** | **IN** | 27.90±23.16 | 29.94±29.69 | 28.16±29.50 | 33.20±28.20 | 33.89±29.06 |
| | **OUT** | -25.97±9.82 | -28.03± 18.58 | -26.24 ± 25.96 | -31.28 ± 31.26 | -31.96 ± 32.38 |
| | **NET** | 1.93 ± 31.64 | 1.91 ± 45.64 | 1.92 ± 52.40 | 1.92 ± 55.50 | 1.93 ± 57.05 |
| | **Vertical transfer** | 2.0 ± 9.61 | | -1.78 ± 9.06 | 5.30 ± 22.98 | 0.68 ± 8.92 |
| | **Recirculation flow** | -2.28 ± 3.38 | | -4.27 ± 6.53 | 3.95 ± 9.02 | 0.59 ± 8.88 |


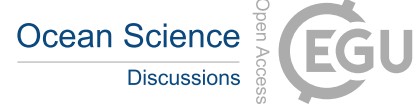

**Table 5. As Table 4 but using monthly data.**

|  |  | G. Cadiz | Espartel | Camarinal Sill | Tarifa Narrows | Alboran |
|---|---|---|---|---|---|---|
| **Water Transport (Sv)** | **IN** | 0.68 ± 0.09 | 0.63 ± 0.09 | 0.48 ± 0.10 | 0.61 ± 0.11 | 0.62 ± 0.13 |
|  | **OUT** | -0.61 ± 0.05 | -0.55 ± 0.05 | -0.41 ± 0.05 | -0.54 ± 0.05 | -0.55 ± 0.06 |
|  | **NET** | 0.07 ± 0.11 | 0.07 ± 0.11 | 0.07 ± 0.11 | 0.07 ± 0.11 | 0.07 ± 0.11 |
|  | **Vertical transfer / Recirculation** | -0.06 ± 0.02 | | -0.14 ± 0.02 | 0.13 ± 0.02 | 0.00 ± 0.02 |
| **Heat Flux (W/m²)** | **IN** | 18.83 ± 3.32 | 17.44± 3.33 | 13.92 ± 3.38 | 16.73 ± 3.65 | 16.87 ± 4.02 |
|  | **OUT** | -13.83 ± 1.75 | -12.39 ± 1.52 | -8.80 ± 1.29 | -11.74 ± 1.49 | -11.82 ± 1.67 |
|  | **NET** | 5.00 ± 3.36 | 5.05 ± 3.39 | 5.12 ± 3.41 | 4.99 ± 3.40 | 5.04 ± 3.42 |
|  | **Vertical transfer / Recirculation** | -1.44 ± 0.45 | | -3.59 ± 0.44 | 2.94 ± 0.54 | 0.08 ± 0.48 |
| **Salt Transport (10⁶ kg·s⁻¹)** | **IN** | 22.23 ± 8.58 | 20.40± 7.99 | 15.75 ± 6.57 | 20.17± 8.21 | 20.31± 8.56 |
|  | **OUT** | -20.59 ± 7.67 | -18.80 ± 6.99 | -14.16 ± 5.41 | -18.59±6.97 | -18.73 ± 7.15 |
|  | **NET** | 1.64 ± 3.68 | 1.60 ± 3.69 | 1.59 ± 3.69 | 1.58 ± 3.69 | 1.58 ± 3.68 |
|  | **Vertical transfer / Recirculation** | -1.79 ± 0.89 | | -4.63 ± 1.73 | 4.43 ± 1.79 | 0.14 ± 0.78 |



**Table 6. As Table 4 but using monthly data from the run that does not include tidal forcing.**

| | | G. Cadiz | Espartel | Camarinal Sill | Tarifa Narrows | Alboran |
|---|---|---|---|---|---|---|
| **Water Transport (Sv)** | **IN** | 0.76±0.05 | 0.69±0.04 | 0.65±0.05 | 0.64±0.05 | 0.67±0.05 |
| | **OUT** | -0.68±0.05 | -0.61±0.05 | -0.57±0.05 | -0.56±0.05 | -0.59±0.05 |
| | **NET** | 0.08±0.03 | 0.08±0.03 | 0.08±0.03 | 0.08±0.03 | 0.08±0.03 |
| | **Vertical transfer / Recirculation** | -0.07 ± 0.02 | | -0.05 ± 0.01 | -0.00 ± 0.01 | 0.02 ± 0.01 |
| **Heat Flux (W/m²)** | **IN** | 20.77 ± 2.83 | 19.19 ± 2.59 | 18.03 ± 2.61 | 17.95 ± 2.60 | 18.47 ± 2.70 |
| | **OUT** | -15.10 ± 1.83 | -13.46 ± 1.64 | -12.29 ± 1.61 | -12.24 ± 1.65 | -12.70 ± 1.78 |
| | **NET** | 5.67±1.63 | 5.73±1.66 | 5.74±1.67 | 5.71±1.67 | 5.76±1.69 |
| | **Vertical transfer / Recirculation** | -1.64±0.42 | | -1.17±0.19 | -0.05±0.10 | 0.46±0.18 |
| **Salt Transport (10⁶ kg·s⁻¹)** | **IN** | 24.24 ± 8.93 | 22.10 ± 8.12 | 20.61 ± 7.62 | 20.57 ± 7.62 | 21.30 ± 7.92 |
| | **OUT** | -22.70 ± 8.42 | -20.55 ± 7.65 | -19.06 ± 7.16 | -19.01 ± 7.16 | -19.74 ± 7.48 |
| | **NET** | 1.54 ± 1.17 | 1.54 ± 1.18 | 1.54 ± 1.18 | 1.56 ± 1.19 | 1.56 ± 1.19 |
| | **Vertical transfer / Recirculation** | -2.15 ± 0.97 | | -1.49 ± 0.59 | -0.05 ± 0.17 | 0.72 ± 0.40 |



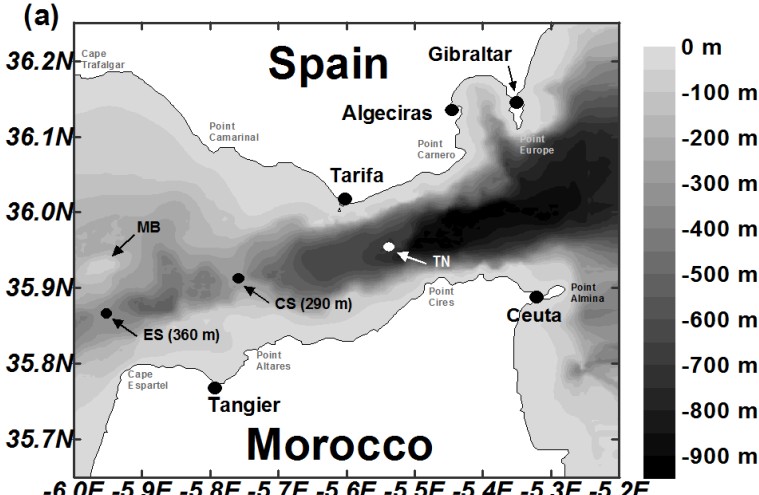

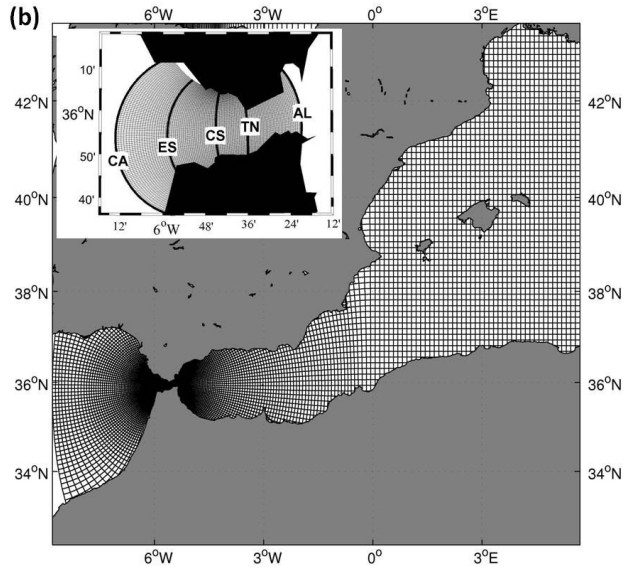

**Figure 1: (a) Map of the Strait of Gibraltar showing the main topographic features: Espartel Sill (ES), Camarinal Sill (CS) and Tarifa Narrows (TN). MB indicates the submarine ridge of Majuan Bank, which divides the Espartel section into two channels: the main channel to the south and a secondary one to the north. (b) Representation of the model grid with the refinement around the Gibraltar Strait. The inset shows the five cross-Strait sections investigated: Gulf of Cadiz (CA), Espartel (ES), Camarinal Sill (CS), Tarifa Narrows (TN) and Alboran Sea (AL).**



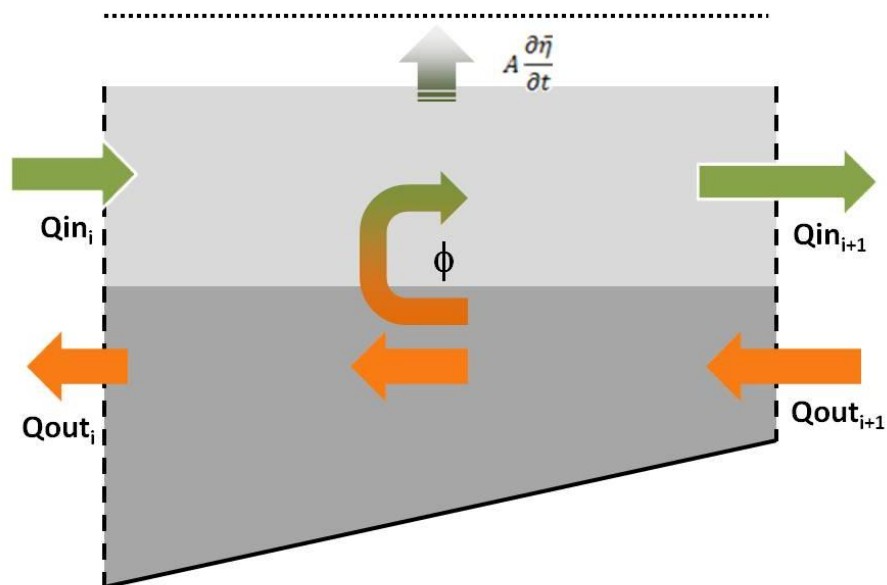

**Figure 2. Sketch of the volume control used to estimate the vertical transfers (Φ) of water between the inflow and outflow layer.
$Q_{in}$ and $Q_{out}$ denote the horizontal transport toward the Mediterranean and toward the Atlantic Ocean at two cross-Strait sections,
(i) and (i+1). $A\frac{\partial \bar{\eta}}{\partial t}$ represents the inflow layer volume change associated with vertical displacements of the mean sea surface
elevation in between the two sections.**





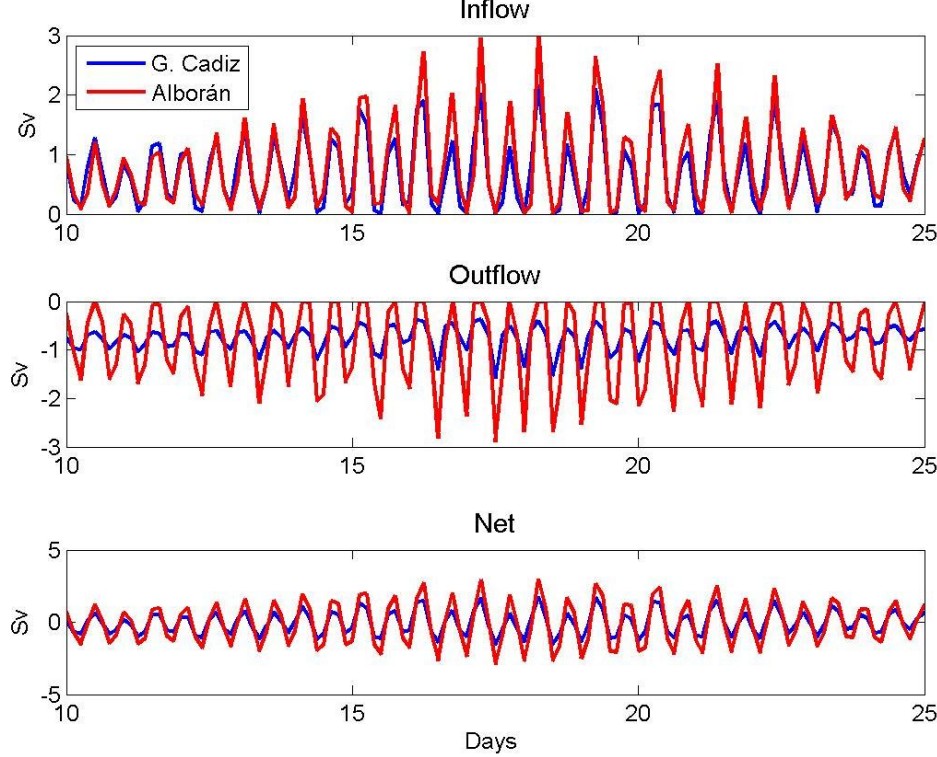

**Figure 3. 3-hourly transports at the outer sections, Gulf of Cadiz (westernmost section, in blue) and Alboran (easternmost section, in red). The inflow (top), outflow (middle) and net transport (bottom) are shown. Only 15 days have been represented for clarity.**





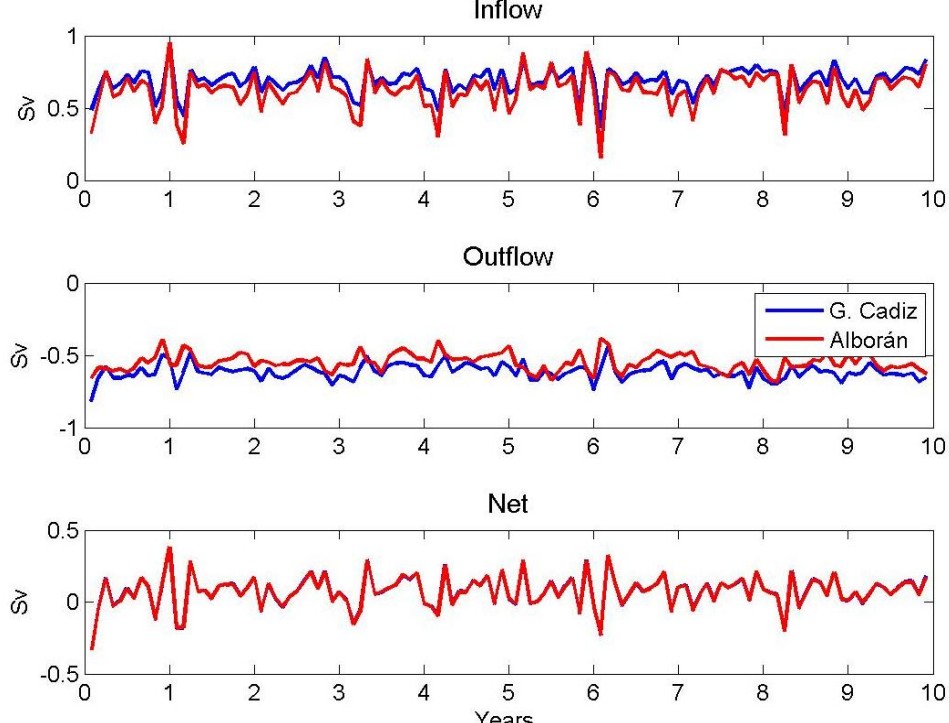

5   **Figure 4. Monthly averages of the 3-hourly transports at the outer sections, Gulf of Cadiz (blue) and Alborán (red). The inflow (top), outflow (middle) and net transport (bottom) are shown.**



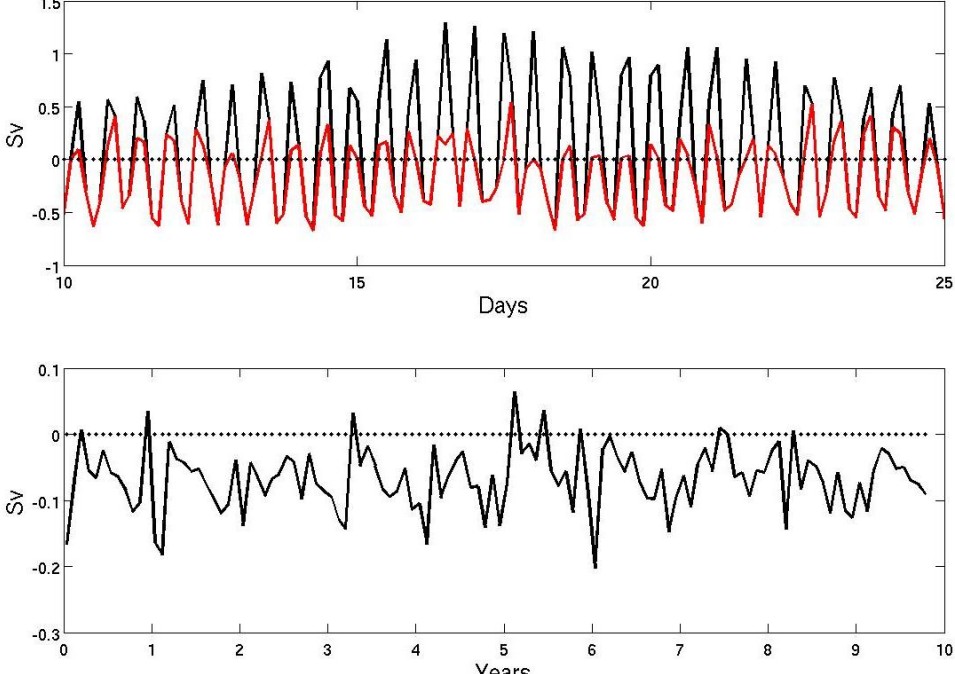

**Figure 5. Upper panel: 3-hourly vertical transfers (black line) between the inflow and outflow layers computed as the difference in the outflow transport between the Gulf of Cadiz and the Alboran Sea. The effective recirculation of water between layers (red line) is also plotted. Positive values indicate transport of water from the outgoing layer to the incoming layer (only a 15 day period is shown for clarity). Lower panel: low frequency transfer/recirculation computed from monthly averages of the transport. Note that the vertical and horizontal scales are different in both plots.**





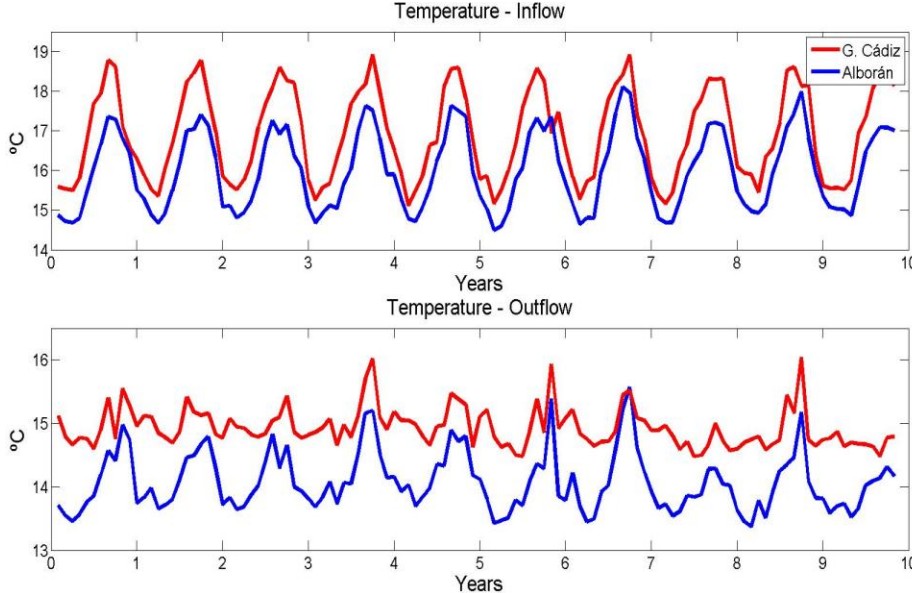

**Figure 6. Monthly time series of temperature at the boundaries of the Strait for (upper panel) the inflow and (lower panel) the outflow.**





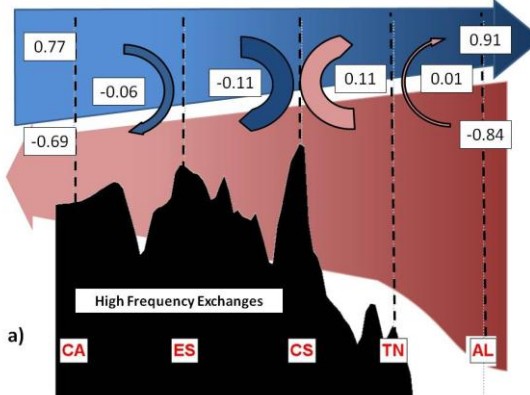

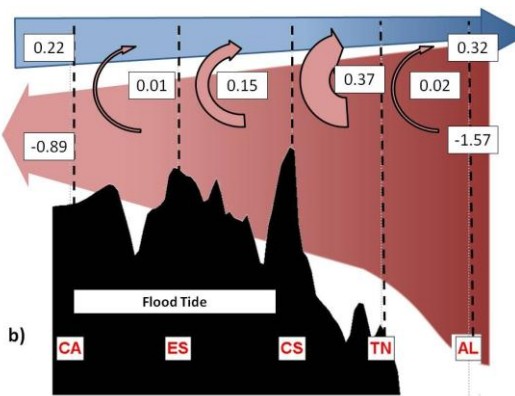

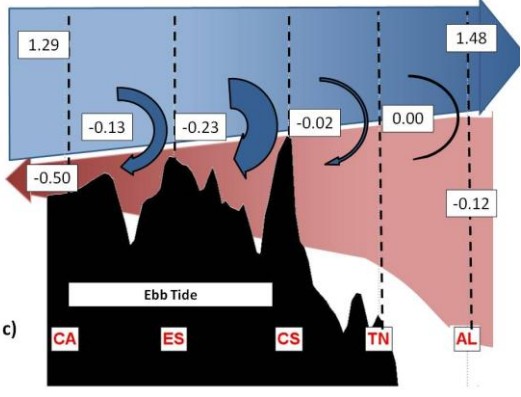

**Figure 7. Two-layer sketch to explain tidal transport divergence in each layer at high frequency (a) Illustration of the mean exchange at high frequency. (b) The exchanges during the flood tide (barotropic tidal current toward the Atlantic Ocean). (c) The same for the ebb tide (tidal current toward the Mediterranean).**




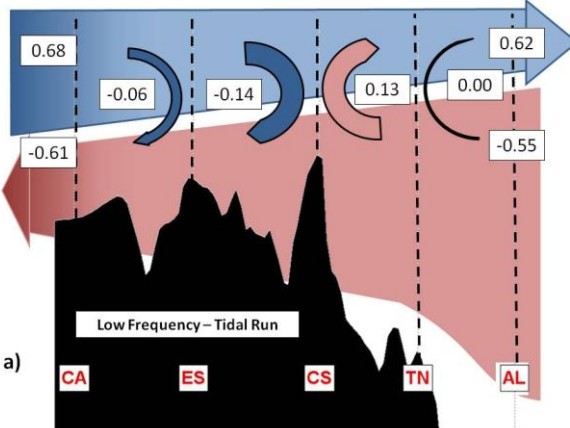

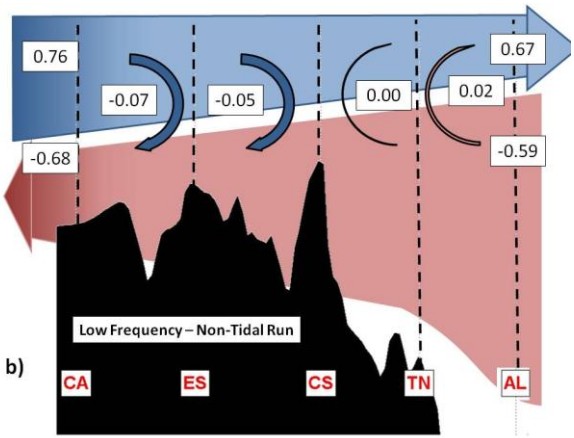

**Figure 8. Two-layer sketch to explain the low-frequency transport divergence associated with tides in each layer. (a) Mean exchange obtained for the tidal run (b) Mean exchange for the non-tidal** run.




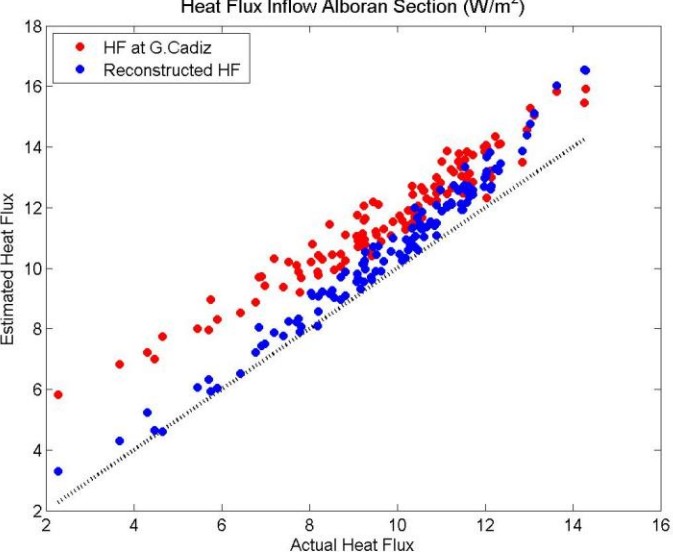

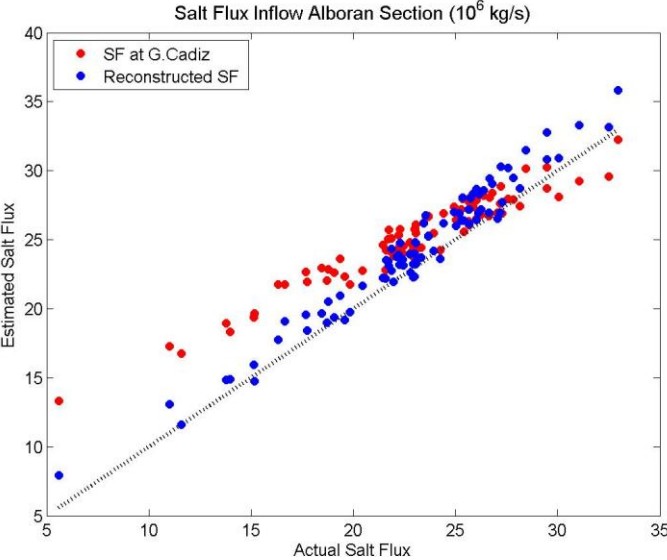

**Figure 9: Comparison between the reconstructed inflow of heat (top) and salt (bottom) at the Alboran section with the actual fluxes (blue dots). The values of fluxes at the western section (red dots) are also included. See text for details.**