# Peer review of "Modelling study of transformations of the exchange flows along the Strait of Gibraltar"

_Ocean Science, 2018_

## Referee Comment (RC1) · Anonymous Referee #1 · 8 Aug 2018

1) This study differs from all the previous studies of the exchange flow in the strait of Gibraltar as it does not consider a given, material salinity surface as the separation of the usual two ( or three ) layers with which the exchange flow is modeled 2) Furthermore, the flow is analyzed not in a single section in the strait but in five successive sections which span the strait all along its length, from the Alboran sea inside the Mediterranean to the Gulf of Cadiz in the Atlantic 3) the numerical model used is the best numerical model developed by Dr. Sannino existing for the Mediterranean. It is forced not only by wind stress, atmospheric pressure and heat/fresh water fluxes at the surface, but also by realistic barotropic tides.The tidal forcing includes both the tide generating potential as a body force and the tidal velocities imposed as lateral

boundary conditions at the open Atlantic boundary. Furthermore, it is the only model which can resolve the hydraulic control occurring at three sections in the Straits thanks to the very high resolution there reached ( 1/200 degree). 4) the very important result obtained is that most of the transformations of water properties along the strait are induced by vertical advection of heat and salt, and not by vertical mixing. The turbulent vertical mixing is what is assumed in coarse resolution models to produce these water masses transformations. In this study, thanks to the very high resolution, vertical mixing is shown to have very little influence. 5) the figures are excellent and make the patterns of vertical exchanges very clear (figs 7-8) One substantial comment and suggestions for improvement 1) I do not believe the explanation given at the top of page 16 for the recirculation of water from one layer to the other in the absence of tidal forcing. The explanation is that mechanical drag would explain the recirculation processes. This might be true in the hypothesis that a material surface separates the two layers, which is not true in this study. Furthermore, the exchanges inside the straits are so small to be almost insignificant. 2) sections 4.1 and 4.2 could be shortened and synthetized. 3) Equations 10 through 15 are cumbersome and distract the reader. I recommend putting them in an appendix and leave only a synthetic explanation in the main text

---

## Referee Comment (RC2) · Anonymous Referee #2 · 3 Oct 2018

The manuscript deals with the modeling reproduction of the Gibraltar Strait fluxes exchange. The authors investigate and quantify the vertical transfer of properties between the inflowing and outflowing waters using a velocity criterion, not done before. Moreover one interesting evidence is the establishment of the limited action of mixing processes, compared with vertical advection forces. There is an interesting approach to the characterization of processes in order also to provide tools for a more realistic implementation of coarser models that parameterize the vertical processes tuning mixing. The manuscript jointly investigate the physical processes affecting the water exchange at the Gibraltar Strait and provide suggestion on the use of the main results for operational modeling. The immediate support to a large modeling community is

evident and despite some doubts on the overall reasoning presented in the paper, it represents an added value for researchers in oceanography.

The paper is linked to a large literature that tackled in the past the same topic, it appropriately quote it, probably sometimes relying on previous evidences a little bit too much (i.e. the reference to Sannino et al 2015 and the results of that paper are mentioned too broadly and there are parts of the text in which the reader is not sure whether some info (on validation, for example) come from the present manuscript or are mentioned from that one. In the discussion section there are two approaches described on how to translate the paper results to improve modeling of the strait for, as example, climate runs. The first one, that link the recirculation flux to net fluxes and provide a relation between them is supported by evidences (fig.9), the other one, just drafted in the last rows of the discussion (from line 9 page 19 on), seem more a speculation and some doubts on the opportunity to mention it are raised.

The paper is structured in a coherent way, even if can be slightly improved. It is written in a good English. The number of tables, probably, can be reduced (suggestion: keep in the same table Tab.1 and Tab.4, where some info are repeated). It has to be clarified the level of generality of the 15 days graphs shown in Figs. 3 and 5a because they are discussed mentioning the max-min values but not clarifying if they represent a typical behavior of the 10 simulated years or not. As a general comment, evidences from figures are discussed, not always introducing to the reader what is shown in that figure. Therefore the reader sometimes jump from one figure to the other but is not helped by the text (example Fig.8). An overall check of this aspect should be done throughout the paper. Even if there is the need to improve the clarity of the presentation of the work done, being more systematic in the presentation of methods (runs done, differences in setups), introducing figures before describing their evidences, I consider these more formal than substantial changes, therefore I suggest to consider the paper for publication after minor revision. Specific comments to support this suggestion are listed below.

Specific Comments:

- Page 3, line 23: since there is an interest in vertical processes, faster than convection, is the choice of a hydrostatic version of the model suitable for investigation? If so, please infer on the added (or not) value of a proper reproduction of the non-hydrostatic component for these specific dynamics. This comment considers that the MITgcm allows also this option, if needed.

- Section 2.1: I would expect in this section just the description of methods and simulation setups. Why not to keep a separate subsection for validation information, probably the first of Results section, instead of mentioning it here, mixed with methodological aspects? It is quite hard to fully understand to what extent the model implementation of Sannino et al., 2015 was the basis of this study, what are the new runs, what are the differences, what was considered validation done in that previous paper and how much is directly validated here. Therefore the request is to dig into the section and try to clarify these points.

- Page 4, line 11. Probably 1/16° resolution for the majority of the Mediterranean basin, increasing it in the strait, is sufficient for a correct reproduction of tidal dynamics and, more generally, of circulation. However, I would appreciate a comment, or references to other works dealing with different, variable resolution applications, inferring the effect of resolution on process reproduction.

- Pag. 4, line 14: it is stated that the vertical discretization is in variable thickness layers, from 3 m on the surface, to 300 m at the bottom. Given the specific focus of the present work, the reproduction of bottom layers with 300 m thickness, to reproduce correctly bathymetry and the dynamics linked to the hydraulic control, is appropriate? How is the last layer set? Variable thickness for the last layer in order to reproduce the correct bathymetry? Please spend some words on this aspect.

- Page 4, line 28. There is the mention to Stanev et al., 2000. Is there the possibility to add info on more recent findings connected with the topic, considering, for example the

work presented by Stanev et al., 2017 (Cascading ocean basins: numerical simulations of the circulation and interbasin exchange in the Azov-Black-Marmara-Mediterranean Seas system- OCEAN DYNAMICS)

- Page 4, from line 31 to the end of section: this part mixes the description of datasets with validation aspects. Does this mean that validation is just mentioned but done in other papers, like for tidal signal in Sannino et al., 2015 or are there aspects directly validated with the new runs (i.e. temperature and salinity)? Going through the paper, am I right saying that three are the runs performed, the first with 3 hourly forcing, the second with monthly mean forcing and a third without tide (the one just mentioned in the discussion and in figure 8 that should be described, as well in the methods, I guess)? Or just the 3 hourly run is done for this paper and the others were accessible from available datasets? Please, help the reader in understanding these points, clarifying and perhaps splitting subsection 2.1.

- Page 5, line 18: why to choose such a not recent period for these simulations? Certainly there is a reason that should be explicitly stated, because the reader would ask why not to consider a recent, well documented by measured data period.

- Page 5, line 26-26 " however, the vertical ...of the basin" this sentence need to be proved.

- Fig. 3 and 5a: clarify why to choose 15 days, if they are a real period taken from the dataset or an average and what are the tidal condition in that period.

- Page 9 , line 11: 2.5 Sv: is that computed as max value on the period or does it refer to fig.3?

- Page 12, lines 10-17: I would move these sentences in the discussion section. There are also other parts, in the Results section that mix the plain presentation of results with the discussion. It could be fair but this, in my opinion, sometimes affects the clarity of results presentation.
- Table 4: in Table 1 and 4 there are some repeated info. I would suggest either to merge the two tables or to express the info just once.

- Page 14, line 10: somewhere you should describe what we see in Fig.8, from what simulations, before discussing them.

- Page 15, line 30: the mention and the description of the non-tidal run setup should be added in the methods section.

- Page 17, line 26: net water fluxes from simulation data as well?

Technical Corrections:

- Page 1, line 27: 250 m. is this value correct? Shouldn't be the CS the shallower point of the strait? Is this a typo?

- Page 13, lines 24-27: some problems with this sentence. Any word missing?

---

## Author Response (AR1)

Firstly, we wish to thank the reviewer for his/her comments to our paper.

Detailed response to comments of referee#1:

**Reviewer comment**: I do not believe the explanation given at the top of page 16 for the recirculation of water from one layer to the other in the absence of tidal forcing. The explanation is that mechanical drag would explain the recirculation processes. This might be true in the hypothesis that a material surface separates the two layers, which is not true in this study. Furthermore, the exchanges inside the straits are so small to be almost insignificant.

Response: We respectfully disagree with the reviewer. It is not required to have a material surface separating the two layers to have a drag acting between the layers. The differences in velocity between the two layers induce a transfer of momentum that is equivalent to a drag force. This happens at tidal frequencies but also at low frequencies. We cannot univocally demonstrate our hypothesis but we do not have an alternative one, that is why we present it as a "possible explanation".

**Reviewer comment**: sections 4.1 and 4.2 could be shortened and synthetized.

Response: we honestly consider the discussion as the most important part of the paper, since it is where we compare our results with previous papers based on the traditional approach of a material surface separating Atlantic and Mediterranean waters. We are therefore reluctant to shorten sections 4.1 and 4.2; actually, subsection 4.1 has been enlarged in the new version of the manuscript, following the suggestions of reviewer#2 (she/he suggested to move some paragraphs and sentences from the results section to the discussion).

**Reviewer comment**: Equations 10 through 15 are cumbersome and distract the reader. I recommend putting them in an appendix and leave only a synthetic explanation in the main text.

Response: we thank the reviewer for her/his comments. However, we think that equations 10 to 15 are important to understand the procedure followed to parameterize the exchanges basing on the reconstruction of the recirculation fluxes in a first step, and subsequently on the estimation of the modification of the T/S properties of the incoming and outgoing waters at the eastern boundary of the Strait. Summarizing, we honestly think that the formulation should be kept in the main body of the manuscript.

Detailed response to comments of referee#2:

We thank the reviewer for her/his comments, which have been useful to improve the manuscript. Below we have responded to each of the specific comments, hoping that these clarifications and amendments meet his/her approval.

**Reviewer comment**: The paper is linked to a large literature that tackled in the past the same topic, it appropriately quote it, probably sometimes relying on previous evidences a little bit too much (i.e. the reference to Sannino et al 2015 and the results of that paper are mentioned too broadly and there are parts of the text in which the reader is not sure whether some info (on validation, for example) come from the present manuscript or are mentioned from that one.

**Response**: the simulation runs used in this paper are those described in Sannino et al. (2015). We have better explained this fact in the new version of the manuscript. As a consequence, we refer to that paper when describing the model configuration and the validation of the tidal amplitude and phase done by Sannino et al. (2015). This has been clarified in the manuscript in order to avoid confusion with the validation of the 3D tidal currents performed here and the results obtained (see comment below).

**Reviewer comment**: In the discussion section there are two approaches described on how to translate the paper results to improve modeling of the strait for, as example, climate runs. The first one, that link the recirculation flux to net fluxes and provide a relation between them is supported by evidences (fig.9), the other one, just drafted in the last rows of the discussion (from line 9 page 19 on), seem more a speculation and some doubts on the opportunity to mention it are raised.

**Response**: in section 4.3 we first present the empirical relationships based on model outputs that can be used to compute, in a simple way, what should be the transformation of water properties along the Strait. This could be used to assess whether the coarse resolution model is representing in a realistic way the water transformation. However, as those models have not enough spatial resolution is very difficult for them to do that job. The only way is to parameterize somehow that process. Therefore, in the second part of section 4.3 we propose a simple way to "force" the coarse resolution models to transform the water properties in their path along the Strait.

**Reviewer comment**: The number of tables, probably, can be reduced (suggestion: keep in the same table Tab.1 and Tab.4, where some info are repeated). It has to be clarified the level of generality of the 15 days graphs shown in Figs. 3 and 5a because they are discussed mentioning the max-min values but not clarifying if they represent a typical behavior of the 10 simulated years or not. As a general comment, evidences from figures are discussed, not always introducing to the reader what is shown in that figure. Therefore the reader sometimes jump from one figure to the other but is not helped by the text (example Fig.8). An overall check of this aspect should be done throughout the paper.

**Response**: Most of these general comments have been raised in the specific comments so we provide the clarifications and amendments below. Related to the last general comment, we have included an introduction to all the figures in the new version in order to clarify what is shown in each of them.

**Reviewer comment:** Page 3, line 23: since there is an interest in vertical processes, faster than convection, is the choice of a hydrostatic version of the model suitable for investigation? If so, please infer on the added (or not) value of a proper reproduction of the non-hydrostatic component for these specific dynamics. This comment considers that the MITgcm allows also this option, if needed.

**Response**: Sannino et al. (2014) compared two twin simulations reproducing the Strait of Gibraltar dynamics differing only for the hydrostatic/non-hydrostatic formulation. When comparing the two model outputs, it was found that the main difference between the two simulations was the presence of an eastward propagating bore in the non-hydrostatic simulation. On the contrary, no relevant differences were found in the simulated hydraulic control, volume transport and tracers vertical profiles. Considering that the non-hydrostatic simulation increases a lot the computational time (about 8 times), and that not relevant differences were found by Sannino et al. 2014, we decided to use the hydrostatic formulation in the model configuration used in this work, and not to rerun the model in its non-hydrostatic version.

**Reviewer comment:** Section 2.1: I would expect in this section just the description of methods and simulation setups. Why not to keep a separate subsection for validation information, probably the first of Results section, instead of mentioning it here, mixed with methodological aspects? It is quite hard to fully understand to what extent the model implementation of Sannino et al., 2015 was the basis of this study, what are the new runs, what are the differences, what was considered validation done in that previous paper and how much is directly validated here. Therefore the request is to dig into the section and try to clarify these points.

**Response**: we have now clarified in the manuscript that we use the model implementation and the two model runs described in Sannino et al. (2015). Moreover, we have included a new subsection within the model description to give the details of the simulation setups. Finally, the validation of the model has been moved to the first subsection of the results as suggested by the reviewer; we have made more clear that the validation of the tidal amplitude and phase was carried out by Sannino et al. (2015), while in our work we have performed a new model validation for the 3D tidal currents, namely at the Espartel and Camarinal sills.

**Reviewer comment:** Page 4, line 11. Probably 1/16° resolution for the majority of the Mediterranean basin, increasing it in the strait, is sufficient for a correct reproduction of tidal dynamics and, more generally, of circulation. However, I would appreciate a comment, or references to other works dealing with different, variable resolution applications, inferring the effect of resolution on process reproduction.

**Response**: 1/16° is now a standard for Mediterranean modelling. For instance, most of the MedCORDEX climate simulations are run at 1/12° (www.medcordex.eu) while the CMEMS official operational product for the Mediterranean is run at 1/24° (http://marine.copernicus.eu/services-portfolio/access-to-products/?option=com_csw&view=details&product_id=MEDSEA_ANALYSIS_FORECAST_PHY_006_013) . Thus 1/16° seems a reasonable option. Moreover, as we focus on the local processes at the Strait of Gibraltar, the influence of the dynamics inside the basin is minor. The most important issue is to have an appropriate resolution around the Strait of Gibraltar, and there we reach 500m following the suggestions of Sannino et al. (2014). We have modified that paragraph as follows to clarify this point:

"The grid has a nonuniform horizontal spacing: over most of the model domain it is 1/16° x 1/16° , which is a standard in present state-of-the-art Mediterranean climate simulations (e.g. www.medcordex.eu). In between the Alboran Sea and the Gulf of Cadiz, following the recommendation of Sannino et al. (2014), the resolution increases up to a maximum value of 1/200° (~500 m) at the Strait of Gibraltar (Fig. 1b)."

**Reviewer comment:** Pag. 4, line 14: it is stated that the vertical discretization is in variable thickness layers, from 3 m on the surface, to 300 m at the bottom. Given the specific focus of the present work, the reproduction of bottom layers with 300 m thickness, to reproduce correctly bathymetry and the dynamics linked to the hydraulic control, is appropriate? How is the last layer set? Variable thickness for the last layer in order to reproduce the correct bathymetry? Please spend some words on this aspect.

**Response**: the model used in this study is a z-level model with partial cell on the bottom. The 300 m vertical resolution refers to the maximum model depth (about 5000 m). Within the Strait the maximum depth does not exceed 800m, and so the vertical resolution on the bottom of the Strait of Gibraltar is only a few tens of meters. We agree with the referee that a better explanation on the vertical resolution is needed. We have added the following text in the new version:

"The layer thickness ranges from 3 m at the sea surface to 300 m at the maximum model depth (~ 5000 m). The partial cell formulation is used for the near-bottom level. Thus, the thickness of the bottom layer will vary according to the bathymetry. The Strait of Gibraltar has a maximum depth that does not exceed 800 m, and the vertical resolution there is only of a few tens of meters. This allows the model to properly reproduce the dynamics linked to the hydraulic control of the exchange flows. "

**Reviewer comment:** Page 4, line 28. There is the mention to Stanev et al., 2000. Is there the possibility to add info on more recent findings connected with the topic, considering, for example the work presented by Stanev et al., 2017 (Cascading ocean basins: numerical simulations of the circulation and interbasin exchange in the Azov-Black-Marmara-Mediterranean Seas system- OCEAN DYNAMICS)

**Response**: in the model configuration described in Sannino et al. (2015), the Black Sea net flow through the Dardanelles Strait was imposed according to the results described in Stanev et al. (2000). The more recent findings on the Black Sea net flow reported by Stanev et al. (2017) have not been used in the two hindcast numerical simulations used here. The geographical scope of this paper is the Strait of Gibraltar and not the Black Sea or the easternmost part of the Mediterranean basin. Thus, we honestly consider that further information on this topic is not particularly relevant for our work and, moreover, it could add confusion to the reader.

**Reviewer comment:** Page 4, from line 31 to the end of section: this part mixes the description of datasets with validation aspects. Does this mean that validation is just mentioned but done in other papers, like for tidal signal in Sannino et al., 2015 or are there aspects directly validated with the new runs (i.e. temperature and salinity)? Going through the paper, am I right saying that three are the runs performed, the first with 3 hourly forcing, the second with monthly mean forcing and a third without tide (the one just mentioned in the discussion and in figure 8 that should be described, as well in the methods, I guess)? Or just the 3 hourly run is done for this paper and the others were

accessible from available datasets? Please, help the reader in understanding these points, clarifying and perhaps splitting subsection 2.1.

**Response**: we thank the reviewer for drawing our attention to this issue. As stated above, the validation of the model has been moved to the results section to avoid any confusion. We have also clarified that the 2D validation of the tidal amplitude and phase was carried out in Sannino et al. (2015), whilst here we have conducted the 3D validation of tidal currents at the main sill of Camarinal and at the secondary sill of Espartel (see the response to a previous comment). Regarding the different runs, in this study we have used two runs, both with 3-hourly data outputs, and only differing in the inclusion ('tidal run') or not ('non-tidal run') of the tidal forcing. These runs are those described in Sannino et al. (2015) and in the revised version of our work they are described in a new subsection (simulation setups) of section 2.1 for the sake of completeness. Additionally, we have computed monthly-mean values from the 3-hourly original outputs (i.e., both for the tidal and the non-tidal runs). This has been highlighted in the new version, in order to avoid confusion.

**Reviewer comment:** Page 5, line 18: why to choose such a not recent period for these simulations? Certainly there is a reason that should be explicitly stated, because the reader would ask why not to consider a recent, well documented by measured data period.

**Response**: The original idea was to simulate the entire ERA-Interim period. However, the simulation was stopped after few years due to the unavailability of the HPC cluster. Moreover, as we focus on describing the mechanisms acting on the Strait at high frequency, the period considered is unimportant.

**Reviewer comment:** Page 5, line 26-26 " however, the vertical ...of the basin" this sentence need to be proved.

**Response**: what we mean here is that the salinity drift is really small compared with the sharp salinity gradients observed at Gibraltar. Thus, a long-term salinity drift of less $o(10^{-3}$ psu/yr) will certainly affect climate studies, but it can hardly affect the recirculation of water in a region (the Strait) where water masses differ in more than 2 salinity units.

**Reviewer comment:** Fig. 3 and 5a: clarify why to choose 15 days, if they are a real period taken from the dataset or an average and what are the tidal condition in that period.

**Response**: in the new version we have clarified that the fifteen days showed in Figures 3 and 5a cover a spring tide subset of the 10-year time series. We chose that period because in a spring tide period it is easier to display the discrepancies between the incoming (outcoming) flows at the outer sections of the Strait. We have added the following sentences to the new version of the manuscript:

"Figure 3 presents an example of the 3-hourly tidal run; namely, it shows the exchange flows computed at the outer limits of the Strait during a 15-days period covering a typical spring tide (days 16 to 22). The spring tide period has been chosen to better display the discrepancies between the inflow and the outflow computed at both sections"; and: "The upper panel of Fig. 5 displays an example of the vertical water transfer $\phi$ computed from the tidal run; the 15-days time period is the same one displayed in Fig. 3."

**Reviewer comment:** Page 9 , line 11: 2.5 Sv: is that computed as max value on the period or does it refer to fig.3?

**Response**: this value has been computed over the 10-year time series. It has been clarified in the new version.

**Reviewer comment:** Page 12, lines 10-17: I would move these sentences in the discussion section. There are also other parts, in the Results section that mix the plain presentation of results with the discussion. It could be fair but this, in my opinion, sometimes affects the clarity of results presentation.

**Response**: we thank the reviewer for drawing our attention to this. The sentence has been moved to the last paragraph of section 4.1, that is, within the discussion section.

**Reviewer comment:** Table 4: in Table 1 and 4 there are some repeated info. I would suggest either to merge the two tables or to express the info just once.

**Response**: merging the two tables is a difficult task due to the amount of information provided in each of them. Also, for completeness we have decided to keep the information in the outer sections in Tables 4 and 5 to ease the interpretation for the reader.

**Reviewer comment:** Page 14, line 10: somewhere you should describe what we see in Fig.8, from what simulations, before discussing them.

**Response**: we have added the following sentence to page 14 of the new version (before the discussion of Fig. 8):
"At low frequency the picture is quite similar. Fig. 8 shows a two-layer sketch to summarize the low-frequency transport divergence associated with tides in each layer. Comparing Fig 8a (tidal run) with Fig 8b (non-tidal run) the role of the tides in the low frequency transports can be assessed (see also Table 5). There is a steady decrease of transports from both the Gulf of Cadiz and Alboran sections to the main sill of Camarinal, where minimum values of the exchange flows are obtained (0.48 Sv and -0.41 Sv for the inflow and outflow, respectively). "

**Reviewer comment:** Page 15, line 30: the mention and the description of the non-tidal run setup should be added in the methods section.

**Response**: in the new version we have added subsection 2.1.1 (simulation setups) to the methods section and included the descriptions of the tidal and non tidal runs.

**Reviewer comment:** Page 17, line 26: net water fluxes from simulation data as well?

**Response**: Yes, if model is well configured, the net transport through Gibraltar will be equal to the net water flux through the sea surface inside the basin.

**Reviewer comment:** Page 1, line 27: 250 m. is this value correct? Shouldn't be the CS the shallower point of the strait? Is this a typo?

**Response**: it is not a typo; the value is correct. The Espartel section has two channels: the secondary northern channel, with a maximum depth of 250m, and the southern main channel, with a maximum depth of 360m. Nevertheless, the reviewer is also right when considering the CS as the shallowest point of the Strait: this assumption is true when considering the main path followed by the exchange flows along the main axis of the Strait. Specifically, in the Espartel section 80% of the Mediterranean outflow flows throughout the southern main channel whilst the remaining 20% flows throughout the secondary northern channel (see e.g. Sánchez-Román et al., 2009). Thus, from the mean exchange point of view one can assume that the Espartel sill is that of the southern channel and has a maximum depth of 360m. On the contrary, at the Camarinal section 100% of the exchange flows throughout its single channel, which has a maximum depth of 290m and is referred to as the Camarinal sill (CS); this is why the CS is considered as the minimum depth for the exchange.

**Reviewer comment:** Page 13, lines 24-27: some problems with this sentence. Any word missing?

**Response**: the reviewer is right. We have reworded the sentence as follows:

"For the tidal induced variability, and from west to east, the mean exchange (see Table 4 and Fig. 7a) increases from the Gulf of Cadiz to the ES and then decreases to a minimum at the CS. East of CS, the exchange increases again reaching a maximum at the easternmost section. This behavior is in good agreement with previous studies (Garcia-Lafuente et al., 2000; Garcia-Lafuente et al.,2013) that already showed stronger tidal currents in the eastern part of the Strait."

Esporles, 30 October 2018

Dear Editor,

please find enclosed the revised version of the manuscript entitled "Modelling study of transformations of the exchange flows along the Strait of Gibraltar" by A. Sánchez-Román, G. Jordà, G. Sannino and D. Gomis to be considered for publication in Ocean Sciences.

We thank the reviewers and editor for their comments, which have been useful improving the manuscript. In the new version we have modified the main text of the manuscript to address the comments and suggestions of the reviewers. In the following there is a list with the added/changed issues:

- We have better described the model configuration and the two runs used in this study; both with 3-hourly data outputs, and only differing in the inclusion ('tidal run') or not ('non-tidal run') of the tidal forcing. These runs are those described in Sannino et al. (2015) and in the revised version of our work they are described in a new subsection (simulation setups) of section 2.1 for the sake of completeness.
- We have clarified that we computed monthly-mean values from the 3-hourly original outputs (i.e., both for the tidal and the non-tidal runs).
- The validation of the model has been moved to the results section to avoid any confusion. We have also clarified that the 2D validation of the tidal amplitude and phase was carried out in Sannino et al. (2015), whilst here we have conducted the 3D validation of tidal currents at the main sill of Camarinal and at the secondary sill of Espartel.
- Reviewer 2 suggested either to merge the tables 1 and 4 or to express the info just once. However, merging the two tables is a difficult task due to the amount of information provided in each of them. Also, for completeness we have decided to keep the information in the outer sections in Tables 4 and 5 to ease the interpretation for the reader.

Yours sincerely,

A. Sánchez Román

[revised manuscript text omitted]

We use the model implementation described in Sannino et al. (2015). In the following there is a brief description of the model

10   configuration: the model uses a curvilinear orthogonal grid covering the entire Mediterranean Sea and part of the Atlantic Ocean, including the Gulf of Cadiz at its western boundary (Sannino et al., 2015). The grid has a nonuniform horizontal spacing: over most of the model domain it is 1/16° x 1/16° , which is a standard in present state-of-the-art Mediterranean climate simulations (e.g. www.medcordex.eu). In between the Alboran Sea and the Gulf of Cadiz, following the recommendation of Sannino et al. (2014), the resolution increases up to a maximum value of 1/200° (~500 m) at the Strait of

15   Gibraltar (Fig. 1b). In the vertical the grid has 72 unevenly spaced z-level in order to adequately resolve the dynamics of the different overlying water masses in the Mediterranean. The layer thickness ranges from 3 m at the sea surface to 300 m at the maximum model depth (~5000 m). The partial cell formulation is used for the near-bottom level. Thus, the thickness of the bottom layer will vary according to the bathymetry. The Strait of Gibraltar has a maximum depth that does not exceed 800 m, and the vertical resolution there is only of a few tens of meters. This allows the model to properly reproduce the dynamics

20   linked to the hydraulic control of the exchange flows. According to Sannino et al. (2015), the model bathymetry was obtained by a merging procedure that involved three different datasets; then a bilinear interpolation on the model grid was applied; and finally, a hand-made check for isolate grid points, islands and narrow passages was conducted. The three datasets used were: the Digital Bathymetric Data Base-Variable Resolution (DBDB) at 1-min resolution for the Mediterranean basin, DBDB-2 (2-min resolution) for the Atlantic box, and the very high resolution digitalized chart of Sanz et al. (1991) for the Strait of

25   Gibraltar. Vertical eddy viscosity and diffusivity coefficients were computed using the turbulence closure model developed by Bougeault and Lacarrere (1989) for the atmosphere and adapted for the oceanic case by Gaspar and Lefevre (1990). The reader is referred to Sannino et al. (2015) for further details about the model description.

2.1.1 Simulation setups

For this study, we use the two hindcast numerical simulations, performed with and without tidal forcing, described in Sannino

30   et al. (2015). According to these authors, initial conditions for temperature and salinity for the two runs were obtained from the Mediterranean Data Archaeology and Rescue (MEDAR) / Mediterranean Hydrological Atlas (MEDATLAS II) database (MEDAR Group, 2002). The model was forced at the surface by the atmospheric pressure, wind stress and the heat and fresh

water fluxes for the period 1958 – 1967 provided by the ECMWF ERA40 reanalysis database (provided by the European Centre for Medium-Range Weather Forecasts), at a temporal resolution of six hours and a spatial resolution of about 1.125° x 1.125°, while the climatological river discharge was prescribed according to Struglia et al. (2004) for the main 68 catchments (Sannino et al., 2015). The Black Sea net flow through the Dardanelles was imposed by following Stanev et al. (2000).

5 The two runs only differ by the inclusion (tidal run) or not (non tidal run) of the tidal forcing. The tidal run includes both the tide generating potential as a body force in the momentum equations, and the lateral boundary condition in the open Atlantic boundary imposed by the tidal velocities produced by the barotropic tidal model of Carrere and Lyard (2003) (Naranjo et al., 2014). The four main lunar, solar and luni-solar ($M_2$, $S_2$, $O_1$, $K_1$) tidal constituents were prescribed (Sannino et al., 2015). Outputs fields form the tidal run were firstly stored at a temporal resolution of 1 day over the entire basin interior. Then this

10 resolution was enhanced to 3 hours in the Strait of Gibraltar area to properly solve the tidal dynamics. Further details of the two model runs can be found in Sannino et al. (2015).

The model outputs used in this study are 3-hourly data from the aforementioned two runs, both spanning a ten years period: from January 1958 to November 1967. The outputs were analysed at five cross-Strait sections to investigate the spatial variability at both tidal and subinertial frequencies (the latter by using monthly-averaged data from the two runs) of the vertical

15 transfers between the incoming and outgoing waters during their passage through the Strait: the two sections located at the boundaries of the Strait (Gulf of Cadiz, CA, and Alboran Section, AL) and the internal sections of Espartel (ES), Camarinal Sill (CS), and Tarifa Narrows (TN; see Fig. 1b). According to results reported by Sannino et al., (2015), the model slightly overestimates the temperature and salinity of the Mediterranean basin. This fact leads to a long-term drift in the Mediterranean salinity (see Fig. 15 in Sannino et al., 2015), which suggests that the model does not reach an equilibrium state concerning the

20 salinity of the whole basin. Thus, the 10-year mean estimates of transports reported in section 3 could be contaminated by this model drift, as they partially depend on the salinity difference between inflowing and outflowing waters; however, the vertical transfers of water, heat and salt between layers at the Strait of Gibraltar do not depend critically on the absolute value of the salinity of the basin. Moreover, because we focus on tidal to monthly frequencies, a small long-term trend is not expected to modify the basic mechanisms analysed here, as it is shown by the validation reported in section 3.1.

25 **2.2 Computation of water, heat and salt horizontal transports**

The exchanges are characterized in the framework of an inflow layer flowing eastward from the Atlantic to the Mediterranean and an outflow layer flowing westward from the Mediterranean to the Atlantic. The volume transport associated with the inflow (outflow) is computed integrating the positive (negative) velocities at a given section:

$$Qin(x,t) = \int_{South}^{North} \int_{bottom}^{\eta} u^+(x,y,z,t)\,dz\,dy, \qquad (1)$$

[revised manuscript text omitted]

**3.1 Model validation**

The validation of the amplitude and phase of modeled tides (2D) through their comparison with observations and previous modelling studies was done by Sannino et al. (2015) through a harmonic analysis (Foreman, 1977) of the simulated sea surface height over the entire Mediterranean basin. To do that, these authors implemented a barotropic experiment in which only the internal and equilibrium tidal forcing were prescribed. They reported that the computed amplitude and the phase of the principal semidiurnal ($M_2$ and $S_2$) and diurnal ($O_1$ and $K_1$) tidal constituents showed in general a good agreement with the tide gauge values reported in the basin while a reasonable agreement was found in the Strait of Gibraltar with amplitudes differing no more than 5 cm for $M_2$ and 7 cm for $S_2$ and deviations in phase around 18° for $M_2$ and 16° for $S_2$.

In this study, we have further validated the model configuration by checking the reliability of 3D tidal currents. To do that, we compared simulated amplitude and phase of the mean velocity vertical profiles at the main sills of ES and CS (see Fig. 1) with in-situ Acoustic Doppler Current Profiler (ADCP) observations collected in the frame of the INGRES Projects (see Sánchez-Román et al., 2008, 2009; and Sammartino et al., 2015 for details). The vertical structure of modeled mean currents in ES and CS (Figure not shown) shows the two-layer character of the flow with an upper layer flowing towards the Mediterranean Sea and a lower layer flowing towards the Atlantic Ocean. They present a general good agreement with the mean profiles obtained from observations exhibiting a correlation coefficient greater than 0.90. The mean depth of the modeled interface between incoming and outgoing waters only differs in 11 m with the one computed from observations in both locations. Furthermore, we found discrepancies in amplitudes lower than 10 cm s$^{-1}$ while deviations in phase lower than 15° were observed.

**3.2 Transport estimates at the boundaries of the Strait**

[revised manuscript text omitted]

**3.3 Overall transformation of the exchange along the Strait**

The difference between the fluxes that cross the Gulf of Cadiz and Alboran sections provides a measure of the vertical recirculation taking place within the Strait. The upper panel of Fig. 5 displays an example of the vertical water transfer $\phi$ of

25 water computed from the tidal run; the 15-days time period is the same one displayed in Fig. 3. Vertical transfers have been estimated as the difference between the outflowing waters measured at the outer sections according to Eq. (1) and the effective recirculation flux $\xi$ computed according to Eq. (6).

The 3-hourly vertical transfer and recirculation of water exhibit large fluctuations (respective STD of 0.63 Sv and 0.38 Sv, Table 1), shifting their sign according to the tidal cycle. Positive values are observed during ebb tides (eastward moving), then

30 suggesting that part of the outflowing waters will be brought towards the inflow layer. Conversely, negative values are obtained during flood tides (westward moving), which implies that a fraction of the inflowing waters will be conveyed towards the outflow layer during this part of the tidal cycle. It is worth noting that only a fraction of the upward vertical transfer results in an effective recirculation flux as defined in Eq. (6); the remainder will contribute to the rising of the sea surface elevation in

[revised manuscript text omitted]
 the inflowing and outflowing waters at high frequencies for the mean exchange (upper panel), the exchanges during a typical flood tide period (barotropic tidal current toward the Atlantic Ocean, panel in the middle), and during a typical ebb tide period (tidal current toward the Mediterranean Sea, lower panel) estimated from the tidal run outputs. For the tidal induced variability, and from west to east, the mean exchange (see Table 4 and Fig. 7a) increases from the Gulf

of Cadiz to the ES and then decreases to a minimum at the CS. East of CS, the exchange increases again reaching a maximum at the easternmost section. This behavior is in good agreement with previous studies (Garcia-Lafuente et al., 2000; Garcia-Lafuente et al., 2013) that already showed stronger tidal currents in the eastern part of the Strait. However, it is worth noting that the exchange at Espartel is larger than at the neighboring sections. This variability in the exchanges implies strong

5 recirculation fluxes of water in between the sections. In particular, between CA and ES the averaged recirculation flux is -0.06 Sv, between ES and CS it is -0.11 Sv, between CS and TN it is +0.11 Sv and between TN and AL +0.01 Sv. East of CS the variability of the vertical recirculation is large (3-hourly STD is ~0.25 Sv everywhere). The pattern of heat and salt advection between layers is consistent with the water recirculation fluxes: the highest vertical transfers of properties are between the sections of ES and CS and, in second place, between CS and TN.

10 Relating the high frequency recirculation fluxes with the phase of the tidal flow provides also an interesting picture. During a typical flood tide (westward barotropic tidal transport, Fig. 7b) the transport in the outflow layer is the highest (-0.89 Sv at CA and -1.57 Sv at AL) and the net vertical recirculation is positive in between all sections (+0.40 Sv in total). The recirculation flux is especially strong between CS and TN (0.37 Sv) and between ES and CS (0.15 Sv). During a typical ebb tide (eastward barotropic tidal transport, Fig. 7c), the transport in the inflow layer is higher (1.29 Sv at CA and 1.48 Sv at AL) and the net

15 vertical recirculation is negative in between all sections (-0.38 Sv in total). In this case, the largest recirculation flux is between ES and CS (-0.23 Sv) and between CA and ES (-0.13 Sv), while it is almost negligible east of CS.

At low frequency the picture is quite similar. Fig. 8 shows a two-layer sketch to summarize the low-frequency transport divergence associated with tides in each layer. Comparing Fig 8a (tidal run) with Fig 8b (non-tidal run) the role of the tides in the low frequency transports can be assessed (see also Table 5). There is a steady decrease of transports from both the Gulf of

20 Cadiz and Alboran sections to the main sill of Camarinal, where minimum values of the exchange flows are obtained (0.48 Sv and -0.41 Sv for the inflow and outflow, respectively). Therefore, west of CS there is a net vertical recirculation flux towards the outflow layer while east of CS the net vertical transfer is positive and contributes to increase the water exchange. This behavior was reported by Garcia-Lafuente et al. (2000) east of CS for both the inflow and outflow and, more recently, Garcia-Lafuente et al. (2011) showed the same pattern between ES and CS. The largest recirculation fluxes are found between ES and

25 CS (-0.14 Sv) and between CS and TN (+0.13 Sv) but on average the net vertical recirculation of water along the Strait is negative (-0.07 Sv). The time variability of the recirculation flux is similar everywhere (monthly STD = 0.02 Sv) and except between TN and AL never implies a change in its sign.

**4 Discussion**

**4.1 Forcing of the recirculation fluxes between incoming and outgoing waters**

30 In order to investigate the mechanism behind the recirculation fluxes between layers we first computed the correlation between the high frequency vertical recirculation and the inflow, the outflow and the net transport from the tidal run. The results suggest that the recirculation fluxes are driven by the outflow variability, since the highest correlations are found with the outflow in between all sections. In particular, the correlation of the outflow with the recirculation flux is -0.64, -0.66 and -0.55 between

... [3]

[revised manuscript text omitted]

**Página 14: [3] Eliminado**     **Usuario de Microsoft Office**     **30/10/18 12:26**

At low frequency the picture is quite similar (Table 5 and Fig. 8a). There is a steady decrease of transports from both the Gulf of Cadiz and Alboran sections to the main sill of Camarinal,